# Atmospheric turbulence strength distribution along a propagation path probed by longitudinally structured optical beams

Huibin Zhou [1] ✉, Xinzhou Su [1], Yuxiang Duan[1], Hao Song[1], Kaiheng Zou [1], Runzhou Zhang [1], Haoqian Song[1], Nanzhe Hu[1], Moshe Tur[2] & Alan E. Willner [1,3] ✉

Atmospheric turbulence can cause critical problems in many applications. To effectively avoid or mitigate turbulence, knowledge of turbulence strength at various distances could be of immense value. Due to light-matter interaction, optical beams can probe longitudinal turbulence changes. Unfortunately, previous approaches tended to be limited to relatively short distances or large transceivers. Here, we explore turbulence probing utilizing multiple sequentially transmitted longitudinally structured beams. Each beam is composed of Bessel-Gaussian ($BG_{\ell=0,k_z}$) modes with different $k_z$ values such that a distance-varying beam width is produced, which results in a distance- and turbulence-dependent modal coupling to $\ell \neq 0$ orders. Our simulation shows that this approach has relatively uniform and low errors (<0.3 dB) over a 10-km path with up to 30-dB turbulence-structure-constant variation. We experimentally demonstrate this approach for two emulated turbulence regions (~15-dB variation) with <0.8-dB errors. Compared to previous techniques, our approach can potentially probe longer distances or require smaller transceivers.

The atmosphere is a medium that affects many applications, such as various forms of air flight and the transmission of light waves[1–5]. The atmosphere is considered an inhomogeneous medium with random fluctuations of varying strengths in both time and space[1,2]. Moreover, atmospheric turbulence can seriously affect various applications, including (a) people and aircraft: turbulence can be characterized by random air motion, which has been identified as a leading cause of serious injury to airline passengers and damage to various forms of aircraft[4,5] and (b) communications and imaging: the temperature variations of the atmosphere can cause spatial and temporal changes in the refractive index. These refractive index changes can induce wander and distortion to light waves and significantly degrade free-space

communication links and imaging systems[1,2]. In each of these cases, knowledge of the inhomogeneous spatial distribution of atmospheric turbulence can help in spatially avoiding or actively mitigating the effects of strong-turbulence regions[5–8].

In general, optical beams can serve as probes for the detection of atmospheric turbulence due to light-matter interaction[5–14]. For example, the total accumulated turbulence can be determined by measuring the turbulence-induced beam wavefront distortion at the end of its propagation path[15–17]. However, knowledge of the specific distribution of turbulence strength at various longitudinal z distances could be of immense value for enabling: (a) aerial platforms to avoid local areas of "flight risky" strong turbulence[4,5] and (b) more accurate adaptive

[1]Department of Electrical and Computer Engineering, University of Southern California, Los Angeles, CA 90089, USA. [2]School of Electrical Engineering, Tel Aviv University, Ramat Aviv 69978, Israel. [3]Dornsife Department of Physics & Astronomy, University of Southern California, Los Angeles, CA 90089, USA. ✉e-mail: huibinzh@usc.edu; willner@usc.edu

optics turbulence compensation in communication links and imaging systems[17–19].

Previously, optical approaches have been demonstrated that can measure the distribution of turbulence strength along a path. One technique is based on LIDAR[5,8–10], which detects a transmitted pulse that is backscattered by air. This method requires equipment at only one terminal but is typically limited to a few hundred meters due to the small power in a reflected pulse[7–10]. Consequently, such a short distance may not provide sufficient warning to avoid turbulence. Another technique that can operate over many kilometers is based on transmitting two beams that intersect each other along the path[6,11–14]. Changing the angle between these beams results in their overlap at different z locations. The two beams experience similar turbulence where they overlap, thus enabling an array of receivers to measure the z-dependent turbulence strength. Due to the angular requirement for long distances, however, this multi-aperture approach tends to either: (i) require a dramatically larger receiver size than transmitter size[20], thus making it far too large to deploy, or (ii) have relatively poor performance near the transmitter for the case of the transmitter and receiver being of similar size[14,21]. An over-arching goal would be to have a relatively accurate z-dependent turbulence probing technique that operates over long distances and be of reasonable size.

In this paper, we propose and demonstrate using longitudinally structured beams for probing turbulence strength (i.e., turbulence structure constant, $C_n^2(z)$) along a propagation path using a single pair of Tx/Rx apertures of similar size. Our longitudinally structured beams are superpositions of multiple low-divergence Bessel-Gaussian ($BG_{\ell=0,k_z}$) modes with different longitudinal wavenumbers $k_z$ and an orbital angular momentum (OAM) $\ell=0$ order. We tailor the complex coefficients for different $k_z$ values to generate a z-dependent beam width. Subsequently, we measure the resultant turbulence-induced power coupling among various $\ell$ orders as signatures for retrieving the z-dependent turbulence strength. Since turbulence affects a wider beam more than a narrower beam, the receiver detects relatively weaker effects where a narrow beam interacts with turbulence. By sequentially transmitting different beams with the narrow section located at different z locations, we extract the turbulence-strength distribution when measuring at the receiver the z-dependent BG modal coupling from $\ell=0$ to $\ell\neq0$ orders. By simulating a 10-km path with up to 30-dB difference in $C_n^2$, our approach shows: (i) relatively uniform probing errors[22,23] (~0.1 to ~0.3 dB); and (ii) a trade-off between probing resolution and transmitter aperture size. We also experimentally demonstrate probing of two emulated turbulence regions of ~15 dB variation and <0.8 dB error. Importantly, our approach has the potential to support: (i) much longer distances than lidar and (ii) much smaller size and uniform performance than crossed overlapping beams.

## Results

Different turbulence-induced effects on probe beams can be utilized as signatures to help retrieve turbulence information. In our approach, we measure the turbulence-induced modal power coupling as signatures. Modal power coupling is defined as the coupling of power from the original spatial mode of the transmitted beam to other spatial modes[3,24,25]. The amount of modal coupling is related to the z-dependent turbulence strength and beam width[24,25]. Using a single Tx/Rx aperture pair, we sequentially transmit bursts of different longitudinally structured beams as z-dependent probes. Each beam is a superposition of multiple $\ell=0$ order $BG_{\ell=0,k_z}$ modes with different longitudinal wavenumbers, $k_z$[26]. $BG_{\ell,k_z}$ modes have two spatial indices: (i) $\ell$ is the number of $2\pi$ phase shifts in the azimuthal direction (i.e., the beam's OAM value), and (ii) $k_z$ is related to the radial wavenumber $k_r$, which determines the radial ring spacings in the intensity profile[27,28]. $k_z$ and $k_r$ satisfy $k_z^2+k_r^2=\left(\frac{2\pi}{\lambda}\right)^2$, where $\lambda$ is the wavelength[27,28]. By designing the complex coefficients of $k_z$ of the $\ell=0$ order BG modes,

the beam width of each beam can be designed to be z-dependent. This z-varying beam width can result in different z- and turbulence-dependent power coupling from the $\ell=0$ BG modes to other $\ell$ orders. Based on measured $\ell$ modal coupling and $k_z$ designed beam widths, we can extract the inhomogeneously distributed turbulence strength along the propagation path. Importantly, instead of other modal basis (e.g., Laguerre-Gaussian modes), we choose combinations of BG modes for creating the z-dependent probe beams to achieve a lower beam divergence[27,28], thus enabling a smaller size Rx aperture.

Detailed principles of our approach using turbulence-induced $\ell$-based power coupling and $k_z$-based longitudinally structured beams are described below.

### Turbulence-induced depletion of power in a launched mode

Optical beams can be distorted when propagating through turbulence, giving rise to power coupling from the transmitted spatial mode into other modes[3,24,25,29]. In our approach, we consider power coupling among $\ell$ modes to characterize such turbulence-induced modal coupling[30]. Applied to a launched fundamental Gaussian beam ($\ell=0$) propagating through a (statistically) longitudinally uniform atmospheric turbulence, the normalized average power remaining at the end of the link on the $\ell=0$ mode, $P(\ell=0)$, as well as a parameter $\beta$, are[24]:

$$P(\ell=0) \approx (I_0(\beta)+I_1(\beta))\exp(-\beta) \qquad (1)$$

and[24]

$$\beta = 1.8025(D)^{\frac{5}{3}}r_0^{-\frac{5}{3}}, \qquad (2)$$

where $I_n(\cdot)$ is the modified $n$th-order Bessel function of the first kind, $D$ is the beam width (second-order definition[31]) and $\beta$ is related to the integration of the turbulence strength along z, as characterized by the Fried parameter, $r_0$[1,3,32]:

$$r_0 = \left(0.423k^2\int_0^L C_n^2(z)dz\right)^{-\frac{3}{5}} \qquad (3)$$

Here, $C_n^2(z)$ is the z-dependent refractive index turbulence structure parameter, $k=2\pi/\lambda$ and $L$ is the total propagation distance. Equations (1–3) indicate that modal coupling increases with increasing turbulence strength (i.e., smaller $r_0$ or larger $C_n^2(z)$) as well as the beam width (i.e., larger $D$). This can be understood as follows: (i) the Fried parameter $r_0$ is a measure of the transverse distance scale after which the turbulence refractive index becomes uncorrelated and (ii) under a given $r_0$, larger beams experience more uncorrelated refractive index fluctuation and exhibit stronger turbulence-induced distortion and modal coupling[24,25].

Heuristically extending Eq. (1) to an inhomogeneous turbulence scenario and comprising $M$ uniform segments of strengths $\{C_{n,j}^2, j=1,\ldots,M\}$ each of thickness $\Delta z$, we express $\beta$ of Eq. (2) as:

$$\beta \approx 1.8025\sum_{j=1}^{M}\left\{\left[0.423k^2 C_{n,j}^2\Delta z\right]D_j^{5/3}\right\}, \qquad (4)$$

where $\{D_j, j=1,\ldots,M\}$ are the beam widths of each respective segment and assuming for simplicity that the widths do not change within a segment. By sequentially transmitting different beams each having its narrow width at a different longitudinal turbulence segment, we can define a set of equations for solving the corresponding turbulence strengths, $\{C_{n,j}^2, j=1,\ldots,M\}$ along the link.

### Longitudinally structured beams

Longitudinally structured beams can be created by a coherent superposition of multiple co-propagating BG modes with different

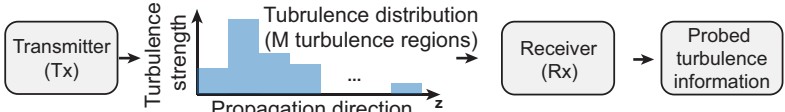

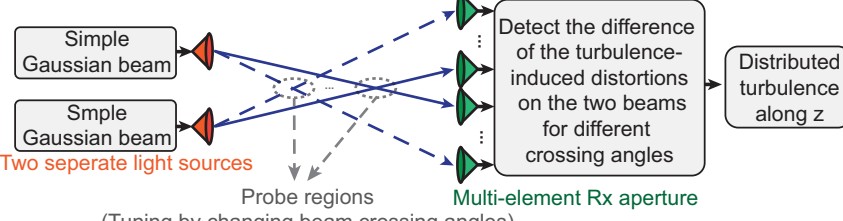

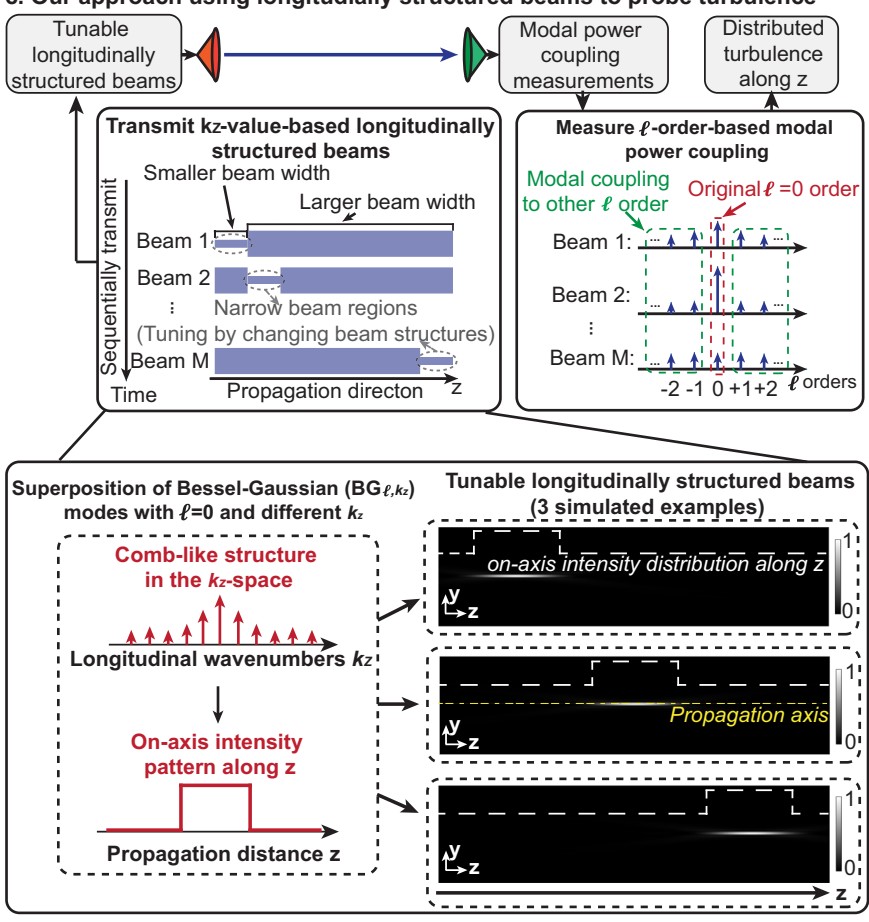

**Fig. 1 | Concept of using longitudinally structured optical beams for probing turbulence along a propagation path. a** A general scheme for transmitting forward-propagating optical beams at the transmitter (Tx) to probe turbulence along a path. At the receiver (Rx), beam-turbulence interactions are measured to retrieve turbulence information. **b** One prior turbulence probing technique transmits two beams from two separate sources, crosses them at different distances, and detects them using a multi-element Rx aperture array. **c** Our approach designs and

sequentially transmits multiple longitudinally structured beams. Our longitudinally structured beams are superpositions of multiple $BG_{\ell=0,k_z}$ modes with different longitudinal wavenumbers $k_z$ and an OAM $\ell=0$ order. The three numerically generated transverse (y)-longitudinal (z) intensity distributions exemplify three different beams with narrower beam widths only in limited regions of choice. At the Rx, based on measured modal coupling from $\ell=0$ to $\ell\neq0$ orders, the distributed turbulence strength along the propagation path is extracted.

longitudinal wavenumbers, $k_z$, and $\ell=0$ orders[26,33–36]. The $k_z$ values have equal spatial frequency spacing, thereby forming a comb in the space-frequency domain[26], as shown in Fig. 1c. The constructive and destructive interference among the BG modes is governed by their complex coefficients. By controlling the coefficients, a longitudinally structured beam can be designed to have a desired on-axis central intensity distribution along the propagation axis z, from z = 0 to L[26,37].

Equation (5) shows the waveform of a longitudinal structured beam consisting of $(2N+1)$ BG modes all at the same optical frequency $\omega_0$[26,38]:

$$\Psi(\rho,z,t) = e^{-i\omega_0 t}G(\rho)\sum_{n=-N}^{N}A_n J_0\left(k_{\rho,n}\rho\right)e^{ik_{z,n}z} \qquad (5)$$

where $\rho$ is the radius in the cylindrical coordinate; $k_{\rho,n}$ and $k_{z,n}$ are transverse and longitudinal wavenumbers, respectively, satisfying $k_{\rho,n}{}^2 + k_{z,n}{}^2 = k^2$; $G(\rho)$ is a Gaussian apodization and is chosen to ensure that all BG modes have relatively low divergence in the relevant longitudinal range[38]; and the coefficients $\{A_n\}$ are given by[26]

$$A_n = \frac{1}{L} \int_0^L F(z) e^{-i\frac{2\pi}{L}nz} dz, \qquad (6)$$

where the function $F(z)$ defines the desired longitudinal on-axis central intensity distribution. Figure 1c shows an example of a longitudinally structured beam that has a rectangular-shaped longitudinal central intensity distribution. Specifically, the on-axis intensity is designed to be higher within $z_i \le z \le z_j$ and lower elsewhere, corresponding to a pattern[26]

$$F(z) = \begin{cases} 1 & \text{for } z_i \le z \le z_j \\ 0 & \text{elsewhere} \end{cases} \qquad (7)$$

Importantly, the location of the intensity-higher region can be almost arbitrarily determined by a proper choice of the $F(z)$ function[33].

As for the z-dependence of the beam width: In the intensity-higher region ($z_i \le z \le z_j$), different BG modes constructively interfere and contribute to a higher intensity at the center part of the beam[27,34], and consequently, a narrow beam width. However, in the other regions, different BG modes are no longer in-phase and their power is spread over a larger width[27,34]. In our turbulence probing approach using longitudinally structured beams, such controllable z-dependent beam width can provide z-dependent signatures that help to extract the turbulence strengths along the z-axis.

## Turbulence probing using longitudinally structured beams

Figure 1b shows a prior turbulence probing approach using two crossing beams. By changing the angle of the two beams, they overlap at different z locations. At these locations, there is a commonality in how the light is affected by turbulence. By detecting the difference in the turbulence-induced distortions for different angles, the turbulence strength along z can be measured. Figure 1c shows the concept of our approach using a single Tx/Rx pair and multiple longitudinally structured beams, each having its narrow width at a different position along the path. Our longitudinally structured beams are superpositions of multiple $BG_{\ell=0,k_z}$ modes with different $k_z$. Proper choice of the superposition coefficients dictates the distribution (along z) of the on-axis intensity pattern of the beam, which also corresponds to different beam widths along z. To simplify the analysis and simulation, the turbulence distribution along z is divided into $M$ equal-length regions, each having constant, region-dependent turbulence strength[6,13,14] (i.e., $\{C_{n,j}^2, j=1,\ldots,M\}$ for the turbulence region j). At the Tx, $M$ longitudinally structured beams are sequentially transmitted, in which beam i has its narrow width in region i. Following Eq. (4), the $\beta$ for beam i can be approximated by:

$$\beta_i \approx 1.8025 \sum_{j=1}^{M} \left\{ \left(D_{i,j}\right)^{\frac{5}{3}} \left[0.423 k^2 C_{n,j}^2 \Delta z\right] \right\} \qquad (8)$$

where $D_{i,j}$ is the width of beam i when it is in turbulence region j, and $\Delta z = L/M$ is the length of each region. We note that the beam width is not a constant value and can change along z during beam propagation within each region[27]. In Eq. (8), we use a constant beam width value $D_{i,j}$ as an approximation to calculate the turbulence effect for each region. For beam i, $D_{i,j}$ is defined as the average value of the beam width along z within turbulence region j. Equation (8) can be

represented as a set of linear equations:1

$$\begin{bmatrix} \beta_1 \\ \vdots \\ \beta_{M-1} \\ \beta_M \end{bmatrix} = \left(1.8025 \times 0.423 k^2 \Delta z\right) \begin{bmatrix} (D_{1,1})^{\frac{5}{3}} & \cdots & (D_{1,M-1})^{\frac{5}{3}} & (D_{1,M})^{\frac{5}{3}} \\ \vdots & \ddots & \vdots & \vdots \\ (D_{M-1,1})^{\frac{5}{3}} & \cdots & (D_{M-1,M-1})^{\frac{5}{3}} & (D_{M-1,M})^{\frac{5}{3}} \\ (D_{M,1})^{\frac{5}{3}} & \cdots & (D_{M,M-1})^{\frac{5}{3}} & (D_{M,M})^{\frac{5}{3}} \end{bmatrix} \begin{bmatrix} C_{n,1}^2 \\ \vdots \\ C_{n,M-1}^2 \\ C_{n,M}^2 \end{bmatrix}$$

$$(9)$$

Based on Eqs. (1–2), the normalized average received power remaining on the $\ell = 0$ order for beam i is approximated by:

$$P_i(\ell=0) \approx \left(I_0\left(\beta_i\right) + I_1\left(\beta_i\right)\right) \exp\left(-\beta_i\right) \qquad (10)$$

Therefore, after measuring $P_i(\ell=0)$ for each transmitted beam at the Rx and extracting the corresponding $\beta_i$ based on Eq. (10), we can solve the $M$ equations of Eq. (9) for the $\{C_{n,j}^2, j=1,\ldots,M\}$ of the different regions.

We note that previous studies have reported a more general analytic expression to calculate normalized average received power on different OAM orders ($P(\ell)$)[24]:

$$P(\ell) \approx \frac{\beta^\ell \cdot {}_2F_2\left(\frac{1}{2}+\ell, 1+\ell; 2+\ell, 1+2\ell, -2\beta\right)}{2^\ell \Gamma(2+\ell)} \qquad (11)$$

where $\Gamma(\cdot)$ is the gamma function and ${}_2F_2$ is the generalized hypergeometric function. Equation (10) is reduced from Eq. (11) for the case of $\ell=0$[24]. Both Eq. (10) and Eq. (11) are approximated formulas that only consider the low-order turbulence aberrations (e.g., tip and tilt)[24]. This approximation is based on that the contribution to OAM modal coupling of higher-order aberrations diminishes rapidly and can be orders of magnitude smaller[24,39]. It has been shown that Eq. (11) generally underestimates $P(\ell)$, but it is in relatively close agreement with the exact values for $\ell=0,1$[24].

In order to measure the normalized $P(\ell=0)$, we perform spatial modal decomposition and normalize the power among OAM orders ranging from $\ell=-10$ to $\ell=+10$[24,27]. If a larger number of OAM orders are taken into account during the measurements, a more accurate normalized value of $P(\ell=0)$ might be obtained. Moreover, it is also possible to utilize the power on many other $\ell \ne 0$ orders as signatures for probing based on Eq. (11). Therefore, more information on the modal coupling can be utilized for turbulence probing with potentially higher accuracy.

## Simulation

To explore the performance of our approach, we simulate the probing of a 10-km path with inhomogeneous turbulence. As an example, in Fig. 2a, we simulate a case of three regions, each with a different constant turbulence strength of $\{C_{n,j}^2, j=1,2,3\}$, respectively. We use a split-step beam propagation method[1,32], in which 40 different phase plates are placed at 250-m spacings. When each beam sequentially propagates through the phase screens, each screen induces its own spatial phase distortion on the beam. We generate "random" phase distributions with different turbulence strengths according to the Kolmogorov turbulence theory[1,32]. To probe three regions, we transmit three longitudinally structured beams (e.g., Beam 1, 2, and 3) at a wavelength of 1550 nm, in which each beam's narrower beam width longitudinally overlaps with its corresponding turbulence region. We set the center of $k_z$ longitudinal wavenumbers to be $Q = k_{z,0} = (1 - 6 \times 10^{-10}) \times k$; $N=7$; and the size of the Tx/Rx aperture diameters = 1 m.

We note that our approach applies Eqs. (1–2) (i.e., equations for a homogeneous medium in ref. 24) to an inhomogeneous turbulence scenario comprising multiple longitudinal segments. In ref. 24, homogeneous turbulence (only a single segment) is considered, and the input beam is a single-mode coherent beam. However, we consider inhomogeneous turbulence cases (multiple segments) in our paper. As

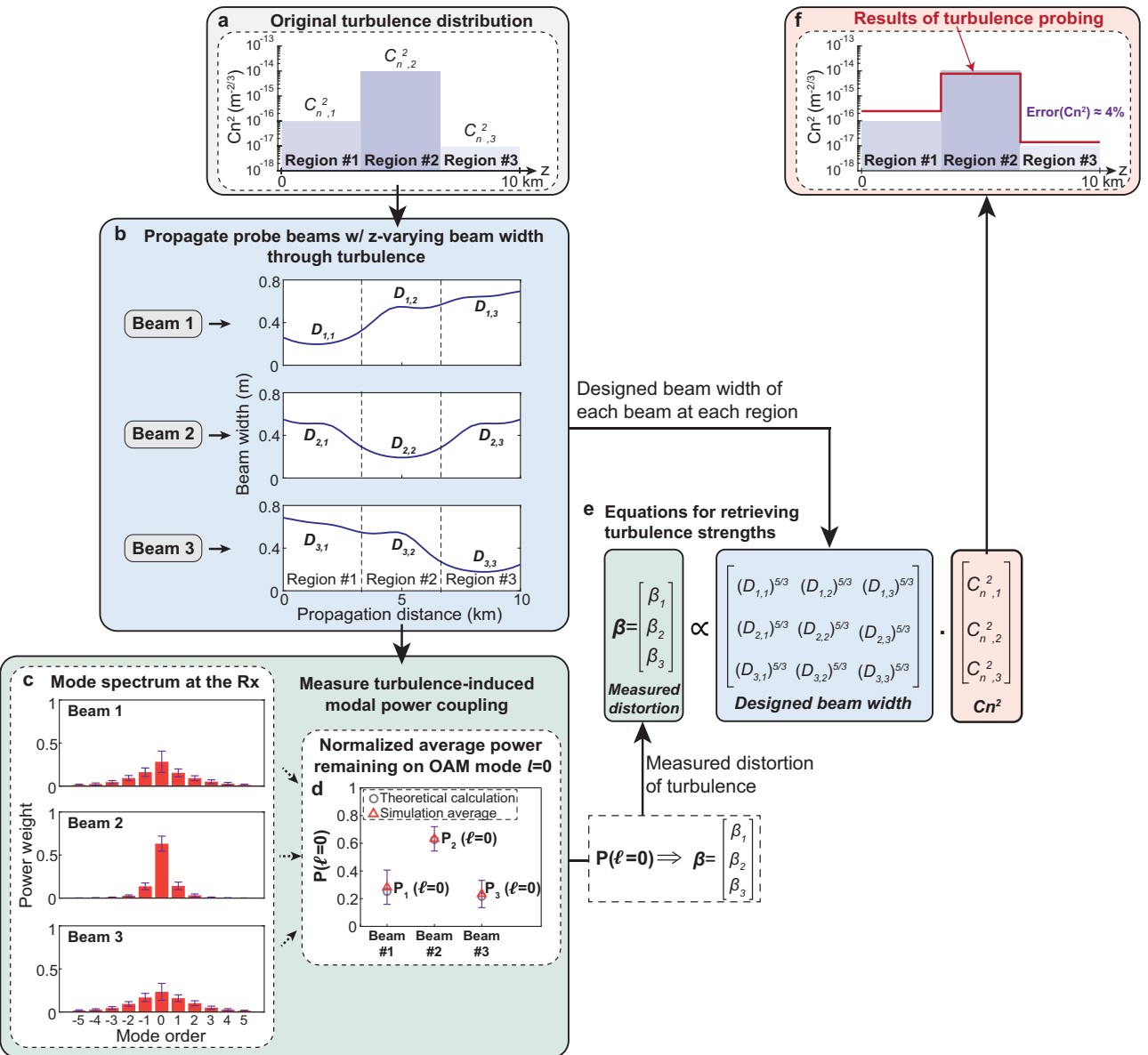

**Fig. 2 | Procedures of our approach for probing a turbulence distribution with three different turbulence regions. a** Three equal-length turbulence regions in simulation with each region having a different turbulence strength of $\{C_{n,j}^2, j=1,2,3\}$. **b** Simulated beam widths ($D_{i,j}$ for each beam i in each region j) of the three beams designed for turbulence probing. **c** Simulated average modal spectrum for the three beams at the Rx under 200 turbulence realizations. The bars show the standard deviation for the simulation results. **d** Normalized average power remained on the $\ell=0$ for the three beams (e.g., $P_i(\ell=0)$ for Beam i) in the simulation. **e** Constructed equations for retrieving the turbulence distribution. **f** Probed turbulence distribution in simulation. The simulated relative probing error is ~4% (~0.2 dB from the original value) for this example of probing three turbulence regions.

a result, the input beam for turbulence regions j (j ≥ 2) can be only partially spatially coherent[40]. This is because the beam can contain many spatial modes after propagating through turbulence, which can generally decrease the spatial coherence of the beam[41,42]. Thus, it is valuable to study whether Eqs. (1–2) are applicable to each beam i propagating through turbulence regions j (j ≥ 2).

In "Supplementary Information" Section 1, we use simulations to help determine whether Eqs. (1–2) can provide reasonable estimations in several cases of inhomogeneous turbulence distributions each comprising three regions. For each case, we simulate $P_{i,j}(\ell=0)$ for beam i at the end of region j and compare it to the theoretically calculated value based on Eqs. (1–2). Our results show that $P_{i,j}(\ell=0)$ become smaller after more turbulence regions (e.g., j becomes larger), which might be due to the more significant accumulation of turbulence-induced modal coupling[42]. The calculated results are in

relatively close agreement with the simulated results, with <5% errors. Our simulation results seem to indicate that Eqs. (1–2) might provide an approximation of turbulence-induced modal coupling results for turbulence regions j (j ≥ 2) in inhomogeneous turbulence cases. However, a more comprehensive and rigorous theoretical analysis may be beneficial in the future in order to further investigate modal coupling effects on partially coherent beams propagating in various inhomogeneous turbulence cases[40–43].

We calculate the beam width $D_{i,j}$ in Fig. 2b for each beam i in each region j, and the width of each beam becomes smaller in its corresponding region (see "Methods" for width calculation). Modal decomposition[27] is used at the receiver to calculate each beam's modal spectrum, and Fig. 2c shows an average of 200 different turbulence realizations (see "Methods" for the decomposition). Subsequently, $P_i(\ell=0)$ for beam i is compared with the theoretical calculation

(Fig. 2d) and used to calculate $\beta_i$ (see Eq. (10)). As a result, equations relating $\beta_i$, $D_{i,j}$, and $C_{n,j}^2$ can be formed (Fig. 2e). Finally, the $C_{n,j}^2$ values are extracted by solving an inverse problem (Eq. (9)) using optimization algorithms (see "*Methods*" for the inverse problem). These simulated probed turbulence strengths (i.e., $C_{n,j}^2(probed)$) are compared to the original values from Fig. 2f. The relative errors of the turbulence distribution's $L^2$ norm[22,23] of our method can be given by:

$$Error\left(C_n^2\right) = \frac{||C_{n,j}^2 - C_{n,j}^2(probed)||_2}{||C_{n,j}^2(probed)||_2} \qquad (12)$$

For this example, the relative error is ~4.0% (-0.2 dB from the original value).

In our approach, we utilize the designed beam width in the vacuum (without turbulence) to form equations for retrieving longitudinal turbulence strengths. However, turbulence can cause beam width variations at a given propagation distance[1,44,45]. Therefore, the actual beam width in turbulence may not be identical to that of our designed beam. Moreover, beam width variations may also affect the location of the intensity-higher region (the smaller beam-width region)[46,47]. In "Supplementary Information" Section 2, we simulate turbulence-induced beam width variations under several turbulence distributions and investigate how such variations would affect our probing approach.

Our results show two effects caused by turbulence-induced beam width variations: (i) the beam width can be affected by turbulence and become larger than the designed beam width in the vacuum (i.e., beam spreading)[44,45] and (ii) the location of the smaller beam-width region can be shifted closer to the transmitter under turbulence (i.e., location shift)[46,47] (see "Supplementary Information" Fig. S2). Moreover, these two effects are related to both the turbulence distribution and the design of the probe beam. Specifically, our results show that (i) the beam spreading tends to be more significant if stronger turbulence is closer to the transmitter[48] and (ii) the location shift is larger for a probe beam with its smaller beam-width region located further from the transmitter. (see "Supplementary Information" Section 2 for more explanations).

Furthermore, we compare the turbulence probing performance when using the designed beam width in the vacuum or the average beam width in turbulence (see "Supplementary Information" Fig. S3). The results show that the probing error is ~2% larger if the beam width changes in turbulence are not considered. We also simulate other turbulence distributions with different numbers of regions and find similar effects of beam width variations on our approach. In these specific cases, our simulation results seem to indicate that our approach may suffer a relatively small decrease in the probing accuracy if we do not consider the beam width changes in turbulence. In general, the beam width variation can become more significant for a beam propagating through a longer and stronger turbulent path[1,44]. Therefore, it may have a greater effect on our approach. We note that a more rigorous theoretical study may be beneficial in the future in order to: (a) examine the extent that beam-width variations affect the accuracy of our approach and (b) help optimize the design of the probe beams for better performance in various turbulence cases[46,47].

Next, we explore more complicated scenarios with 40 turbulence regions. In these examples, we simulate "Gaussian-shaped" turbulence distributions with the peak turbulence strength located at different z (see Fig. 3). We design and transmit 40 longitudinally structured probe beams (Beams 1 to 40). For designing Beam i, we use an on-axis central intensity distribution

$$F_i(z) = \begin{cases} 1 \text{ for } z_i \leq z \leq z_i + 250m \\ 0 \text{ elsewhere} \end{cases} \quad (z_i = (i-1) \times 250m). \qquad (13)$$

Figure 3a shows the received beam profiles of Beams 1, 20, and 40 for a turbulence distribution with its peak strength located at z = 5 km. We find that Beam 20 has its minimum size at z = 5 km and has the least relative distortion of the 40 beams. Figure 3b shows the average $P(\ell = 0)$ under 200 turbulence realizations. The results show that the average $P(\ell = 0)$ is higher for the beams that have narrower widths in the stronger turbulence regions. This is because the turbulence causes relatively less distortion and modal power coupling to narrower beams. Figure 3c1–c5 compare the simulated probed turbulence distributions with the original values. Our results show relatively uniform performance with errors varying from ~2% to ~8% (-0.1 to ~0.3 dB from the original value), for various z locations of peak turbulence strength.

To further investigate the probing resolution of our approach, we vary the width of the Gaussian-shaped turbulence distribution (see Fig. 4). We compare different center longitudinal wavenumbers $Q$ of the longitudinally structured beams, and $Q$ is considered an important property describing longitudinally structured beams[33]. Specifically, the max number of BG modes ($2N_{max} + 1$) that can be used to construct the beams is related to the $Q$ value according to[33]:

$$N_{max} \leq \frac{(k - Q)L}{2\pi} \qquad (14)$$

where $k = 2\pi/\lambda$ and $L$ is the path length. As shown in Fig. 4a, our simulation shows that smaller $Q$ results in the following: (i) there tend to be more rapid changes in the beam width along z, and (ii) the minimum beam width becomes smaller. These results are due to the following: (i) a smaller $Q$ supports a larger $N_{max}$, such that more BG modes can be used to represent the narrower on-axis intensity distribution $F_i(z)$; and (ii) the beam spot size of a BG beam becomes smaller with a smaller longitudinal wavenumber[33, 49].

We simulate the probing performance under three different Gaussian-shaped turbulence distributions, each having a different full width at half maximum (FWHM) (Fig. 4b, c). With a smaller $Q$, the longitudinally structured beams can probe a narrower (i.e., faster changing) Gaussian-shaped turbulence distribution with smaller errors. This is because the beam with a smaller $Q$ has more contrast and sharper beam-width transitions, which can efficiently sense more rapid turbulence changes along z.

Although a smaller $Q$ may have better performance, it also tends to require a larger transmitter aperture. The required transmitter aperture diameter $D_{aperture}$ for generating probe beams is given by[26,33]:

$$D_{aperture} \geq 2L\sqrt{\frac{k^2}{\left(Q - \frac{2\pi N}{L}\right)^2} - 1} \qquad (15)$$

From Eq. (15), a larger aperture is required to generate longitudinally structured beams with a smaller $Q$[26]. Therefore, there tends to be a trade-off between the probing resolution and the transmitter aperture size. One can achieve finer probing resolution by (i) utilizing larger apertures that can support smaller $Q$ or (ii) designing and transmitting more beams such that each beam can probe a shorter-range turbulence region.

In order to explore the performance of our approach in different cases, we simulate other turbulence distributions (see "Supplementary Information" Fig. S4). These results include "linear-changing," "triangular-shaped," and "sine-shaped" distributions with errors of <8, <12, and <32%, respectively. Simulation results show that more complicated distributions with larger longitudinally spatial gradients tend to result in larger errors. We also simulate a $C_n^2$ profile based on the Hufnagel-Valley (H-V) model, which has been widely used to describe turbulence strength at different altitudes[50]. In addition, we simulate our approach and compare the results to previously published experimental data of $C_n^2$ at different altitudes measured by

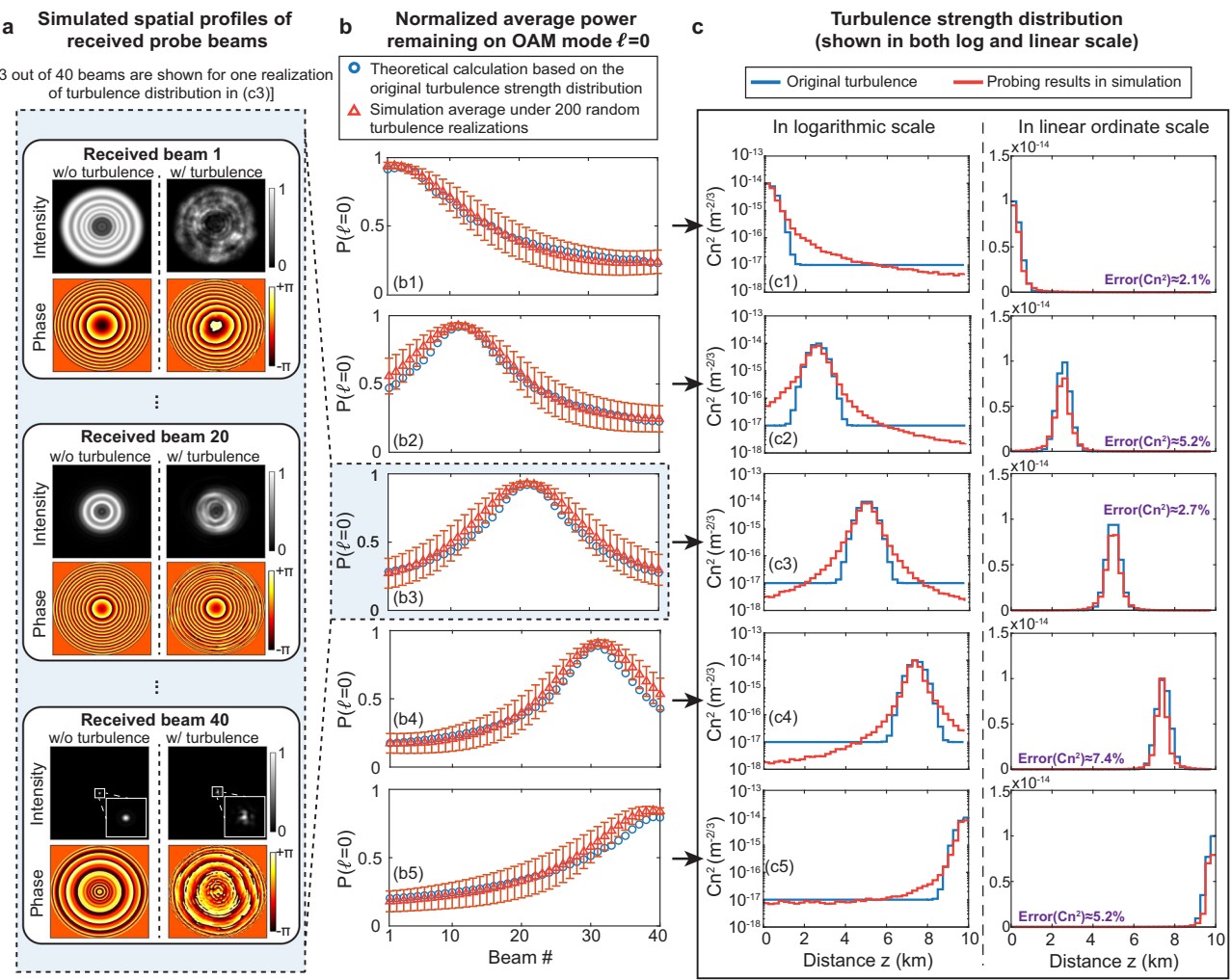

**Fig. 3 | Simulation results for probing "Gaussian-shaped" turbulence distributions, each with the peak turbulence strength located at a different distance, z, in a 10-km path. a** Received beam profiles of Beam 1, 20, and 40 for probing the turbulence distribution with its peak strength located at z = 5 km under one turbulence realization. **b** Average $P(\ell = 0)$ for the probe beams under 200 turbulence realizations. The bars show the standard deviation of the results. **c** The original turbulence distribution ($C_n^2(z)$) and its simulated probed values.

radiometers[51] (see "Supplementary Information" Fig. S4). The simulated probing error compared to the H-V model and the experimental data published in the literature[51] is ~8 and ~16%, respectively.

## Experimental validation

We conduct a proof-of-concept experiment with emulated turbulence regions under laboratory conditions, as shown in Fig. 5a (see the "Methods" for more details). We divide a 0.6-m propagation path into two equal-length regions, and we emulate turbulence effects by placing a rotatable thin phase plate in the middle of each region. The phase plates are fabricated to have different Fried parameter $r_0$ to emulate different turbulence strengths (smaller $r_0$ means stronger turbulence) for a wavelength of 1550 nm[52].

To probe these two turbulence regions, we generate and transmit Beams 1 and 2, having their narrower beam widths at $0 < z < 0.3$ m and $0.3$ m $< z < 0.6$ m, respectively. Figure 5b, c shows the simulated and experimentally measured intensity profiles of the two beams. In Fig. 5d, we calculate the beam widths for the two beams based on the measured intensity profiles. Our results show that the experimentally measured beam widths are in relative agreement with the simulation results. In Fig. 5e, we show the received beam profiles and measured modal spectra of Beams 1 and 2 under one turbulence realization for four different cases, where turbulence regions 1 and 2 have different

values for $r_0$. As shown in the measured modal spectra, the beam suffers less turbulence-induced modal coupling when its narrower-width section is in the stronger turbulence (smaller $r_0$). For example, in Figs. 5e2, 4, Beam 1 exhibits a larger measured $P(\ell = 0)$ than that of Beam 2 for the stronger turbulence region 1 and weaker turbulence region 2 (see Fig. 5e3, e5) for the opposite scenario).

Figure 6a shows measurements of the $P(\ell = 0)$ for Beams 1 and 2 under 200 different turbulence realizations for the four different turbulence distribution cases. Again, the histograms show that the average $P(\ell = 0)$ is larger for the beam that has smaller beam widths in the stronger turbulence region. Compared to Case 1 & 2 with $r_0 = 1$ mm and 3 mm, Cases 3 & 4 with $r_0 = 0.4$ mm and 3 mm result in more mode coupling to $\ell \neq 0$ orders and smaller $P(\ell = 0)$ due to the stronger turbulence ($r_0 = 0.4$ mm) region. To probe the turbulence strengths in the two regions, we first calculate for each case the average $P(\ell = 0)$ for Beams 1 and 2 for the 200 turbulence realizations. Using Eq. (9), we then utilize the average $P(\ell = 0)$ and measured beam widths (shown in Fig. 5d) to extract the $r_0$ for the two regions. As shown in Fig. 6b, the average relative errors in the experiment for Cases 1 to 4 are around 7.7, 9.8, 9.4, and 15.3% (i.e., all <0.8 dB), respectively. Simulations are also conducted using the same parameter settings, and these results are in relatively good agreement with the experimental results.

**a   Simulated beam width of probe beams with different center longitudinal wavenumber Q (3 out of 40 beams are shown)**

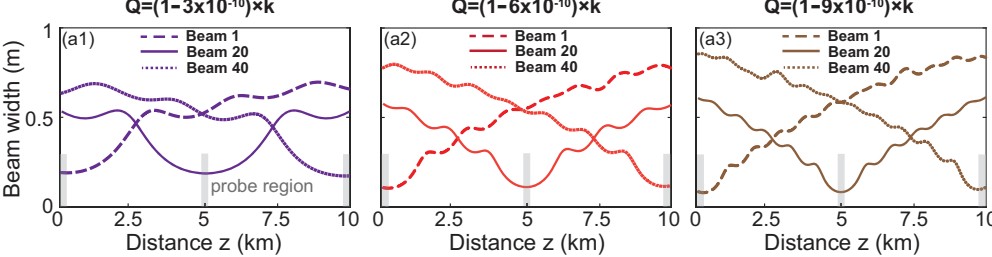

**b   Simulated turbulence probing results using probe beams with different Q for 3 different turbulence distribution cases**

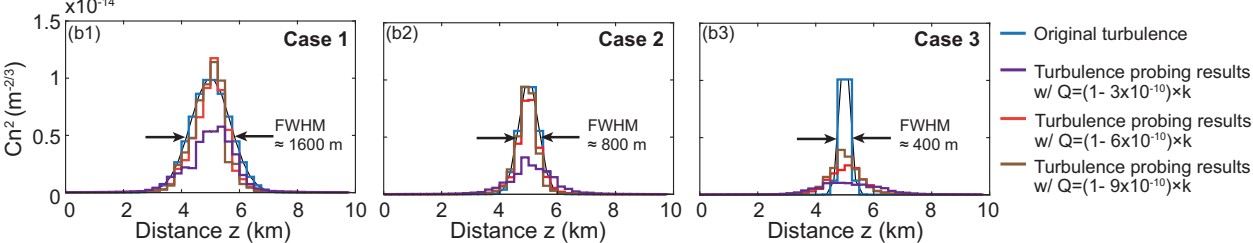

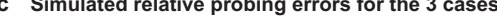

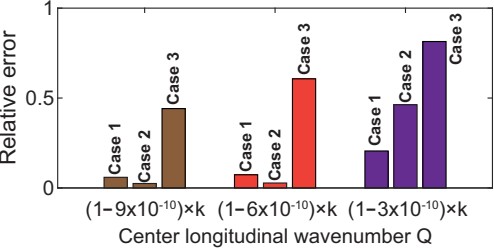

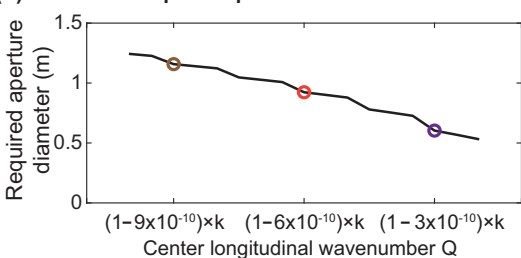

**Fig. 4 | Simulated turbulence probing results by using longitudinally structured beams with different center longitudinal wavenumbers Q. a** Simulated beam widths of the longitudinally structured beams with different center longitudinal wavenumbers (i.e., $Q$ values with wavenumber $k = 2\pi/\lambda$). For the different $Q$ values, we choose $N = N_{max}$. **b** Simulation results for probing "Gaussian-shaped" turbulence distributions with different full widths at half maximum (FWHM). **c** Relative probing errors for the three different cases of FWHM using longitudinally structured beams with different $Q$ values. **d** Calculated aperture diameters required for generating the longitudinally structured beams with different $Q$ values.

## Discussion

Currently deployed, non-optical turbulence probing techniques are limited, including (i) radar-based turbulence monitoring has difficulty measuring turbulence in clear air without dense clouds[5] and (ii) an aircraft flying through high-turbulence locations can signal other aircraft, but this does not help the original aircraft avoid the turbulence or if the turbulence changes dynamically[5,53].

Alternatively, optical probing techniques do not generally suffer from these drawbacks[5–7,13]. However, different optical approaches can have different advantages and disadvantages when compared to the approach of our paper:

(a) Back-scattering-based lidar: Lidar relies on detecting relatively low-power backscattered light by air[5] and is typically limited to a few hundred meters[7–10]. This distance provides a few seconds of warning time, which is typically too short for an airplane to actively avoid turbulence[5,7]. Our approach detects forward-propagating beams and has a much higher signal power, potentially supporting multi-kilometer-length distances and providing a longer warning time for turbulence avoidance. However, since lidar uses equipment at one terminal and our approach requires separate Tx and Rx terminals, lidar can probe in any direction and does not need to form a link between two terminals.

(b) Forward-propagating crossed beams: Crossed beams relies on changing the angle and location of beam overlap[14,21]. It has been demonstrated for long distances, but: (i) the receiver array can require many elements and be very large in size (e.g., 4 m for a 10-km probe) and (ii) the accuracy near the transmitter is generally poorer. Alternatively, our approach utilizes a single pair of Tx/Rx apertures, the size can be smaller (e.g., 4X smaller), and the accuracy can be uniform along the entire path. However, the crossed beams approach uses simple Gaussian beams, which are easier to generate and transmit than our tunable structured beams.

(c) Focusing Gaussian beams: Given our structured beams with a z-dependent width, we compare our approach to a simple case of using transmitter lenses to focus a fundamental Gaussian beam at different distances[54–56]. In the "Supplementary Information", we compare a focused Gaussian beam and our longitudinally structured beams for a 10-km path with a 1-m Tx aperture. Our simulation shows that focused Gaussian beams have: (i) sharper beam width changes and finer probing resolution, but (ii) larger beam divergence resulting in a larger Rx aperture (e.g., ~3x larger receiver aperture when the focus is at 1 km).

Another relevant issue to consider is the system architecture. Our approach requires creating a link between two terminals, which places a limitation on the directions that can be probed. One can envision several scenarios that can reduce the impact of this limitation and create capabilities/opportunities, including the following:

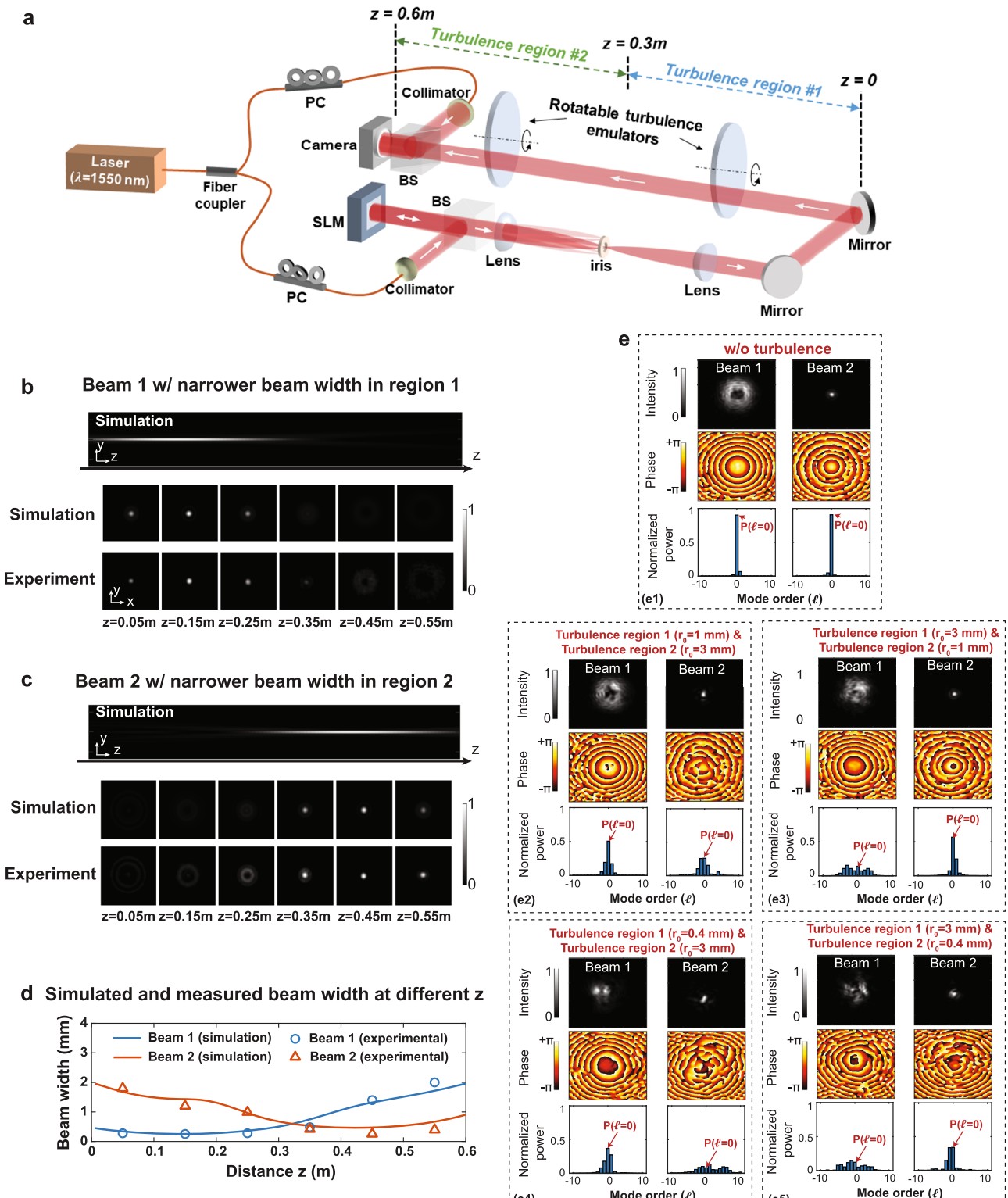

**Fig. 5 | Experimentally measured beam width and modal coupling results for our designed probe beams for probing a turbulence distribution with two different turbulence regions. a** Experimental setup for probing two emulated turbulence regions using two longitudinally structured beams (see "Methods" for more details). In the experiment, we set $Q = 0.999997k$ and $N = 2$ to generate the two beams (Beam 1 and 2). They have narrow beam widths at $0 < z < 0.3$ m and

0.3 m < z < 0.6 m, respectively. Simulated and experimentally measured **b**, **c** intensity profiles and **d** beam widths of Beam 1 and 2 at different propagation distances. **e** Experimentally measured profiles and modal spectra of Beam 1 and 2 under one turbulence realization for four different turbulence cases, where regions 1 and 2 have different $r_0$.

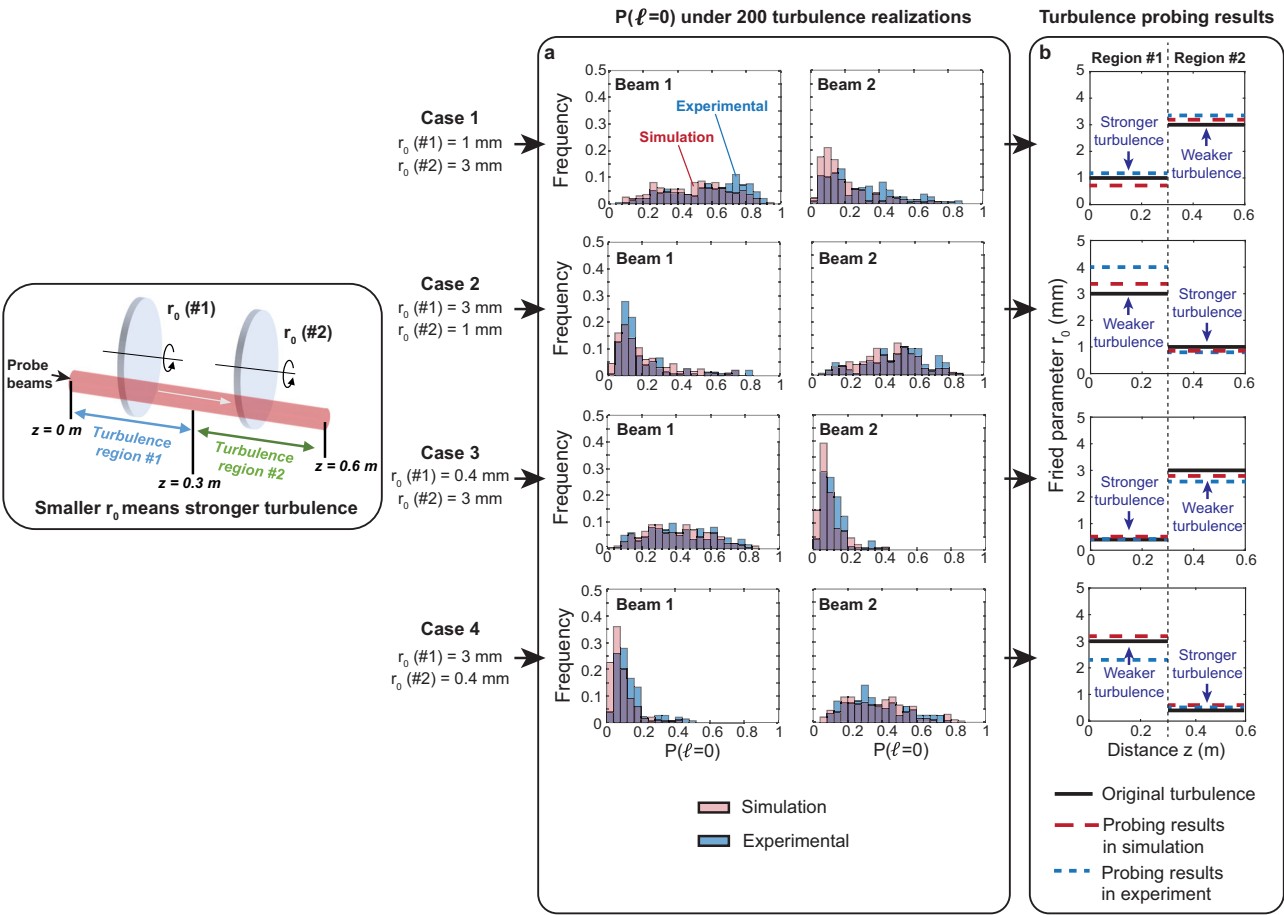

**Fig. 6 | Turbulence-induced modal power coupling under 200 different turbulence realizations and our turbulence probing results in simulation and experiment. a** Simulated and experimentally measured $P(\ell=0)$ for Beam 1 and 2 under 200 turbulence realizations for the four different turbulence distribution cases. **b** original turbulence and probed turbulence in simulation and experiment.

To directly compare probed turbulence strengths with the emulated ones, we show the probing results in terms of the Fried parameters $r_0$ instead of the $C_n^2$. These two parameters are mutually related by Eq. (3). Based on Eq. (3), the equivalent $C_n^2$ of the stronger turbulence region is up to ~30X (~15 dB) larger than that of the weaker one.

(a) Reconfigurable path: One aircraft can dynamically and reconfigurably steer the probe beam to many other aircraft for sequentially probing turbulence in different directions.

(b) Receiver at touchdown: When an aircraft is landing, the receiver terminal can be located on a ground station, thereby enabling an aircraft to probe the turbulence of the landing path.

(c) Network of beams: Multiple probe beams can form a topological grid among different platforms. For example, a network of aircraft, satellites, and ground stations can each contain many transmitters and receivers, thereby enabling a mapping network to probe the 3-dimensional distribution of turbulence in a region of the atmosphere.

## Summary

In this paper, we propose and demonstrate an approach using sequentially transmitted longitudinally structured beams to probe turbulence strength along a propagation path. Each beam is a superposition of multiple BG modes and is designed to have a smaller beam width at a different longitudinal region. Since turbulence can affect a wider beam more than a narrower beam, we extract the turbulence-strength distribution after measuring the turbulence-induced modal coupling for different beams at the receiver. Our simulation shows (i) relatively uniform probing errors (~0.1 to ~0.3 dB) of our approach for a 10-km turbulence path with up to a 30-dB difference in $C_n^2$ and (ii) a trade-off between probing resolution and transmitter aperture size. We experimentally demonstrate our approach for probing two

emulated turbulence regions with up to a ~15-dB turbulence strength variation. The experimental results show <0.8-dB probing errors. Compared to previous methods, our approach has the potential to (i) support longer distances or (ii) achieve fairly uniform performance along the path using smaller transceivers.

## Methods

**Experimental details of our turbulence probing demonstration**
As shown in Fig. 5a, at the Tx, we generate longitudinally structured beams by encoding the desired pattern into a computer-generated hologram on a programmable spatial light modulator (SLM)[49]. A Gaussian beam with a beam waist of ~7 mm is incident on the SLM and acts as the input of the complex amplitude phase modulation for structuring the spatial amplitude and phase of the longitudinally structured beam. Next, a 4-f system and an iris are used to filter out the desired beam and remove the undesired diffraction orders[49]. We divide the 0.6-m path into two equal-length regions and emulate turbulence effects by placing a rotatable thin phase plate in the middle of each region (i.e., 0.3-m regions with at least 0.15-m free-space propagation after the plate). The phase plates are fabricated with a pseudo-random phase distribution obeying Kolmogorov spectrum statistics[1,24,57,58]. They are characterized by different Fried parameters $r_0$ (e.g., 0.4, 1, and 3 mm) to emulate different turbulence strengths; smaller $r_0$ corresponds to stronger turbulence[1,57]. In order to emulate different turbulence realizations, we rotate the phase plates so that the beam passes through different representations of turbulence[57].

We examine if our emulation of turbulence after ≥0.15-m beam propagation after the phase plate can be considered close to Kolmogorov statistics by measuring the Strehl ratio (SR) for a Gaussian beam[57,59,60] (see "Supplementary Information" Fig. S7). We measure SR values for different phase plates under various link lengths (from 0.3 to 0.6 m). Our results show that the measured values are close to the theoretical values with <8% relative errors. These results indicate that the emulated turbulence exhibits a reasonable representation of a Kolmogorov power spectrum. In addition, we measure the statistics of the power fluctuation of the received beam passing through a phase plate with $r_0 = 1$ mm. The probability density function of measured fluctuations follows a lognormal model[57] with a correlation coefficient R > 0.96 to Kolmogorov statistics for various link lengths and propagation after phase plates. Moreover, the scintillation index is found to be larger for a longer link, which indicates larger intensity fluctuations caused by the emulated turbulence[61]. If the receiver is placed right after the phase plate, we note that the received beam has negligible intensity fluctuations and the effective turbulence strength approaches zero[61]; intensity fluctuations would thus arise only after the beam has propagated some distance after the plate[61].

Previously, phase plates have been utilized in various laboratory experiments for emulating turbulence with Kolmogorov statistics in a relatively short path[24,57,59,60]. Importantly, the turbulent path emulated by phase plates in the laboratory can correspond to a much longer path[1,62]. Turbulence parameters of a longer path can be scaled from a shorter lab path given that the two systems have a similar Fresnel number[62]

$$F = a^2/(\lambda L) \tag{16}$$

where a denotes the radius of the source aperture and L is the link length. Based on our calculation, each turbulence region emulated in our experiment corresponds to a 5-km path segment with $C_n^2 = 1.3 \times 10^{-14}$ m$^{-2/3}$, $3.2 \times 10^{-15}$ m$^{-2/3}$, and $4.3 \times 10^{-16}$ m$^{-2/3}$ for phase plates with $r_0 = 0.4$ mm, 1.0 mm, and 3.0 mm, respectively.

To probe these two turbulence regions, we design and simultaneously transmit two longitudinally structured beams (i.e., Beam 1 and 2), having their narrow beam widths at $0 < z < 0.3$ m and $0.3$ m < $z < 0.6$ m, respectively. At the Rx, we use off-axis holography to measure the spatial amplitude and phase of each beam and numerically calculate its modal spectrum and $P(\ell = 0)$[52] (see "Supplementary Information" for the off-axis holography approach). To measure the modal spectrum using this approach, we split the laser light source to provide a coherent reference light beam[52,63]. We note that one can also use an SLM at the Rx to perform the inner product between the received beam and Bessel basis functions for the modal decomposition, which does not require a coherent light beam[27,64]. The modal spectrum results for each beam and turbulence distribution is an average of measurements under 200 different turbulence realizations, achieved through the rotation of the phase plates to different orientations[52]. Based on the measured $P(\ell = 0)$, we perform offline algorithms to retrieve the probed turbulence by solving Eq. (9) (see "Methods" for solving the inverse problem). To directly compare probed turbulence strengths with the emulated ones, we show the probing results in terms of the Fried parameters $r_0$ instead of the $C_n^2$. These two parameters are mutually related by Eq. (3). Based on Eq. (3), the equivalent $C_n^2$ of the stronger turbulence region is up to ~ 30X (-15 dB) larger than that of the weaker one.

## Spatial modal decomposition
To calculate the modal power coupling and the power remaining on the OAM $\ell = 0$, we perform the modal decomposition on the received beam. Since the longitudinally structured beam is a superposition of $2N + 1$ BG modes, it may seem a natural choice to decompose the beam into multiple Bessel basis modes[27]. The modal power coefficients on

each Bessel basis $J_\ell(k_{\rho,n}\rho)e^{i\ell\phi}$ can be obtained by the inner product between the received beam field $U(\rho,\phi,z=L)$ at the Rx and each mode basis[27], i.e.,

$$|g_{n,\ell}|^2 = |\int\int U(\rho,\phi,z=L)\left[J_\ell\left(k_{\rho,n}\rho\right)e^{i\ell\phi}\right]^* \rho d\rho d\phi|^2 \tag{17}$$

where $(\rho,\phi)$ are polar coordinates. In order to calculate the modal spectrum, we sum the coefficients on different wavenumbers $k_{\rho,n}$ for each order $\ell$. As a result, the coefficients for the modal spectrum is

$$|g_\ell|^2 = \sum_{n=-N}^{N} |g_{n,\ell}|^2 \tag{18}$$

and normalized such that $\sum_\ell |g_\ell|^2 = 1$. Therefore, the normalized power remaining on the $\ell = 0$ is $P(\ell = 0) = |g_0|^2$.

## Beam width calculation
For the calculation of the beam width, the second moment of the intensity is usually employed, as given by the following equation[31]:

$$D = 2\sqrt{\frac{2\int\int \rho^2 I(\rho,\phi,z)\rho d\rho d\phi}{\int\int I(\rho,\phi,z)\rho d\rho d\phi}} \tag{19}$$

where $I(\rho,\phi,z) = |U(\rho,\phi,z)|^2$ is the beam intensity profile and $(\rho,\phi)$ are polar coordinates. In this paper, we calculate turbulence-induced modal coupling by decomposing the beam into Bessel basis (i.e., $J_\ell(k_{\rho,n}\rho)e^{i\ell\phi}$) instead of into the pure $2\pi$ azimuthal phase change (i.e., $e^{i\ell\phi}$ without $\rho$-related terms). Therefore, we modify the beam width calculation by considering the additional Bessel function $J_\ell(k_{\rho,n}\rho)$ imposed on the beam. As a result, the beam width defined in this paper is calculated by

$$D = \frac{1}{2N+1}\sum_{n=-N}^{N} 2\sqrt{\frac{2\int\int \rho^2 I_n(\rho,\phi,z)\rho d\rho d\phi}{\int\int I_n(\rho,\phi,z)\rho d\rho d\phi}} \tag{20}$$

where $I_n(\rho,\phi,z) = |U(\rho,\phi,z)J_0(k_{\rho,n}\rho)|^2$

## Retrieve the turbulence by solving the inverse problem
Based on Eq. (9), the $C_n^2$ information in different turbulence regions is retrieved by solving an optimization problem to minimize the relative residual norms[12]. The solution $\boldsymbol{X}$ is determined by

$$C_n^2 = argmin\left\{\frac{||\boldsymbol{MX} - \boldsymbol{\beta}||_2}{||\boldsymbol{\beta}||_2}\right\} \tag{21}$$

Where $\boldsymbol{M} = \left(1.8025 \times 0.423k^2\Delta z\right)\begin{bmatrix} (D_{1,1})^{\frac{5}{3}} & \cdots & (D_{1,M-1})^{\frac{5}{3}} & (D_{1,M})^{\frac{5}{3}} \\ \vdots & \ddots & \vdots & \vdots \\ (D_{M-1,1})^{\frac{5}{3}} & \cdots & (D_{M-1,M-1})^{\frac{5}{3}} & (D_{M-1,M})^{\frac{5}{3}} \\ (D_{M,1})^{\frac{5}{3}} & & (D_{M,M-1})^{\frac{5}{3}} & (D_{M,M})^{\frac{5}{3}} \end{bmatrix}$ and

$\boldsymbol{\beta} = \begin{bmatrix} \beta_1 \\ \vdots \\ \beta_{M-1} \\ \beta_M \end{bmatrix}$.

The MATLAB-constrained nonlinear optimization function *fmincon* was used to find the minimum of Eq. (21)[12].

## Reporting summary
Further information on research design is available in the Nature Portfolio Reporting Summary linked to this article.

## Data availability

All raw data, theory details, and simulation detail that support the findings of this study are available from the corresponding authors upon request. Source data are provided with this paper.

## Code availability

All relevant computing codes that support the findings of this study are available from the corresponding authors upon request.

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

## Acknowledgements

This work is supported by the Vannevar Bush Faculty Fellowship sponsored by the Basic Research Office of the Assistant Secretary of Defense for Research and Engineering and funded by the US Office of Naval Research (N00014-16-1-2813); US Office of Naval Research through a MURI award (N00014-20-1-2558); AFOSR (sub-award of FA9453-20-2-0001); Airbus Institute for Engineering Research; Qualcomm Innovation Fellowship.

## Author contributions

H.Z. and A.E.W. conceived the idea; H.Z., X.S., and Y.D. developed the simulation model and performed the simulation; H.Z., X.S., and A.E.W. designed the experiment; H.Z., X.S., Y.D., Hao Song., and K.Z. conducted the experimental measurements; H.Z., X.S., Y.D., R.Z., Haoqian Song., and N.H. carried out the data analysis; M.T. and A.E.W. provided the technical support. All the authors contributed to the interpretation of the results and manuscript writing. The project was supervised by A.E.W.

## Competing interests

The authors declare no competing interests.
