## [Peer Review File · Nature Communications]

Atmospheric turbulence strength distribution along a propagation path probed by longitudinally structured optical beamsREVIEWER COMMENTS

Reviewer #1 (Remarks to the Author):

Results are interesting but the style of the paper must be improved. There are too many long sentences and captions to the figures which must be truncated.

Some sentences are incorrect

"the atmosphere can seriously affect various applications?" seriously? Maybe turbulence?"

The technique is called LIDAR, this is an acronym.

Caption to Fig 1 must be shortened and a brief description must be moved to the text.

Lines 96-102 too long sentence!!!!!!!!!!!!!!

Formula in 101 is incorrect, check brackets.

.....

My main concern is related to the comparison of theoretical results with the experimental data. I do not believe that at the distance of 0.6 m we have a fully developed turbulence, otherwise that we have Kolmogorov's statistics. But the main results of the theory are based on the Kolmogorov's similarity theory. Is it possible to provide a proof, that turbulence in the lab experiment was fully developed?

Clear conclusions (and conclusions in general) are absent. Please, add.

In the abstract authors claimed

"Here, we explore turbulence probing utilizing multiple

15 sequentially transmitted longitudinally structured beams between a single transmitter/receiver aperture pair. "

The refractive index structure parameter measurements are available in the literature data. Is it possible to provide

a comparison with the available experimental data in atmosphere?

Furthermore, it is not clear what is the role of the orbital angular momentum in calculations, because the formula 10 is the approximate formula. Is it possible to add the discussion about OAM and how it will affect the results and accuracy?

Reviewer #2 (Remarks to the Author):

The authors treated the problem as to probing the distribution of turbulence strength along a path. The basic idea of this manuscript is interesting and the work is important. The approach suggested by the authors may be useful for some application scenarios. However, there are some drawbacks in the current version of the manuscript that deserve careful consideration.

1. As for Eq. (2), is the parameter D the beam width at the transmitting plane or the receiving plane? If D is the beam width at the receiving plane, it will depend on the turbulence strength distribution due to the turbulence-induced beam spreading. In this sense, D should not be independent of r_0 . If D is the beam width at the transmitting plane, the authors should note that the beam wave field at the transmitting plane should be coherent and hence D is actually the beam width of a coherent beam. For Eq. (4), D_j with $j = 2, 3, \dots, M$ therein should be considered as the beam width of a partially coherent beam at the input plane of a segment because the beam becomes partially coherent during the propagation in turbulence. Eq. (4) is based on Eq. (2). For a partially coherent beam, whether Eqs. (1) and (2) are valid or not? If not, the theoretical foundation of the manuscript is problematic. It seems that Ref. [24] did not consider the partial coherence issue.

2. As an optical beam propagates through atmospheric turbulence, the beam width may not be

identical to that of the vacuum propagation case. Indeed, atmospheric turbulence may change the beam width at a given propagation distance. However, in "Beam width calculation" of the "Method" section, the authors did not mention the turbulence-induced beam width variation. In my opinion, the authors should analyze how the turbulence-induced beam width variation affects the accuracy of their approach.

3. Because of the existence of the turbulence-induced beam width variation, the design of the location of intensity-higher region should consider the effects of atmospheric turbulence. I suggest that the authors address this issue rigorously.

4. What does the parameter $k_{r,n}$ in Eq. (5) represent? The authors did not describe the meaning of $k_{r,n}$. However, $k_{p,n}$ was explained in the text that follows Eq. (5).

In my opinion, the author should revise the current manuscript carefully before it can be accepted.

The authors sincerely thank the reviewers for all their insightful and excellent comments. We have listed below all the comments from the reviewers, with each comment followed by our response and changes in the revised manuscript highlighted with ***italics/bold/underlined*** formatting. We hope that the manuscript is now suitable for publication in *Nature Communications*.

Reviewer 1

Comment 1 (a): *Results are interesting but the style of the paper must be improved. There are too many long sentences and captions to the figures which must be truncated.*

Response 1 (a): We appreciate the reviewer for raising these important issues. Specifically, we have truncated many long sentences and figure captions.

Below are several examples of truncated sentences in the main text of the manuscript:

- In the *Abstract*, page 1:

-Sentence before revision

“Each tailored beam is composed of Bessel-Gaussian ($BG_{\ell=0,k_z}$) modes of different k_z values such that a distance-varying beam width is produced, resulting in a distance-dependent and turbulence-dependent BG modal power coupling to $\ell \neq 0$ orders.”

-Sentence after revision

“Each tailored beam is composed of Bessel-Gaussian ($BG_{\ell=0,k_z}$) modes with different k_z values such that a distance-varying beam width is produced. This distance-varying beam width results in a distance- and turbulence-dependent BG modal power coupling to $\ell \neq 0$ orders.”

- In the *Introduction* section, paragraph 1 on page 1:

-Sentence before revision

“The temperature variations of turbulence can cause spatial and temporal changes in the refractive index, thereby causing wander and distortion to lightwaves and significantly degrading free-space communication links and imaging systems [1,2]”

-Sentence after revision

“The temperature variations of the atmosphere can cause spatial and temporal changes in the refractive index. These refractive index changes can induce wander and distortion to light waves and significantly degrade free-space communication links and imaging systems [1,2].”

- In the *Introduction* section, paragraph 3 on page 2:

-Sentence before revision

“One technique is based on lidar [5,8–10], and detects a transmitted pulse that is backscattered by air; this method requires equipment at only one terminal but is typically limited to a few hundred meters due to the small power in a reflected pulse [7–10]; such a short distance may not provide sufficient warning to avoid turbulence.”

-Sentence after revision

“One technique is based on LIDAR [5,8–10], which detects a transmitted pulse that is backscattered by air. This method requires equipment at only one terminal but is typically limited

to a few hundred meters due to the small power in a reflected pulse [7–10]. Consequently, such a short distance may not provide sufficient warning to avoid turbulence.

- In the *Introduction* section, paragraph 4 on page 2:

-Sentence before revision

“We tailor the complex coefficients for different k_z values to generate z -dependent beam width, and subsequently measure the resultant turbulence-induced modal power coupling among various ℓ orders as signatures for retrieving the z -dependent turbulence strength.”

-Sentence after revision

“We tailor the complex coefficients for different k_z values to generate a z -dependent beam width. Subsequently, we measure the resultant turbulence-induced power coupling among various ℓ orders as signatures for retrieving the z -dependent turbulence strength.”

- In the *Concept* section, paragraph 1 on page 5:

-Sentence before revision

“In our approach, we measure the turbulence-induced modal power coupling (i.e., the coupling of power from the original spatial mode of the beam to other spatial modes) as signatures [3,24,25], and such modal coupling is related to the z -dependent turbulence strength and beam width [24,25].”

-Sentence after revision

“In our approach, we measure the turbulence-induced modal power coupling as signatures. The modal power coupling is defined as the coupling of power from the original spatial mode of the transmitted beam to other spatial modes [3,24,25]. The amount of modal coupling is related to the z -dependent turbulence strength and beam width [24,25].”

- In the *Concept* section, paragraph 1 on page 5:

-Sentence before revision

“By designing the complex coefficients of k_z of the $\ell = 0$ order BG modes, the beam width of each beam can be designed to be z -dependent, therefore resulting in different z - and turbulence-dependent power coupling from the $\ell = 0$ BG modes to other ℓ orders.”

-Sentence after revision

“By designing the complex coefficients of k_z of the $\ell = 0$ order BG modes, the beam width of each beam can be designed to be z -dependent. This z -varying beam width can result in different z - and turbulence-dependent power coupling from the $\ell = 0$ BG modes to other ℓ orders.”

- In the *Concept* section, paragraph 1 on page 6:

-Sentence before revision

“Since the constructive and destructive interference among the BG modes is governed by their complex coefficients, a longitudinally structured beam can be designed to have a desired on-axis central intensity distribution along the propagation axis z , from $z=0$ to L [26,37].”

-Sentence after revision

“The constructive and destructive interference among the BG modes is governed by their complex coefficients. By controlling the coefficients, a longitudinally structured beam can be designed to have a desired on-axis central intensity distribution along the propagation axis z , from $z = 0$ to L [26,37].”

- In the *Concept* section, paragraph 1 on page 7:

-Sentence before revision

“Figure 1(c) shows an example of a longitudinally structured beam that has a rectangular-shaped longitudinal central intensity distribution with the on-axis intensity higher within $z_i \leq z \leq z_j$ and lower elsewhere [26].”

-Sentence after revision

“Figure 1(c) shows an example of a longitudinally structured beam that has a rectangular-shaped longitudinal central intensity distribution. Specifically, the on-axis intensity is designed to be higher within $z_i \leq z \leq z_j$ and lower elsewhere [26].”

- In the *Simulation* section, paragraph 2 on page 10:

-Sentence before revision

“Subsequently, the normalized average power remaining on $\ell=0$ for each beam (e.g., $P_i(\ell = 0)$ for beam i) is compared with the theoretical calculation (Fig. 2(d)) and used to calculate β_i (see Eq. (10)), thereby forming the equations relating β_i , $D_{i,j}$, and $C_{n,j}^2$ (Fig. 2(e)).”

-Sentence after revision

“Subsequently, $P_i(\ell = 0)$ for beam i is compared with the theoretical calculation (Fig. 2(d)) and used to calculate β_i (see Eq. (10)). As a result, equations relating β_i , $D_{i,j}$, and $C_{n,j}^2$ can be formed (Fig. 2(e)).”

- In the *Simulation* section, paragraph 1 on page 13:

-Sentence before revision

“The results show that the average $P(\ell = 0)$ is higher for the beams that have narrower widths in the stronger turbulence regions, and this is because the turbulence causes relatively less distortion and modal power coupling to narrower beams.”

-Sentence after revision

“The results show that the average $P(\ell = 0)$ is higher for the beams that have narrower widths in the stronger turbulence regions. This is because the turbulence causes relatively less distortion and modal power coupling to narrower beams.”

- In the *Simulation* section, paragraph 1 on page 15:

-Sentence before revision

“In order to explore the performance of our approach under more general cases, we simulate other turbulence distributions (Fig. S1 in the “Supplementary Information” for simulation results),

including linear, “triangular-shaped”, and “sine-shaped” distributions with errors of <8%, <12% and <32%, respectively.”

-Sentence after revision

“In order to explore the performance of our approach in different cases, we simulate other turbulence distributions (see Fig. S4 in the “Supplementary Information”). These results include “linear-changing,” “triangular-shaped,” and “sine-shaped” distributions with errors of <8%, <12%, and <32%, respectively.”

- In the *Experimental Validation* section, paragraph 2 on page 17:

-Sentence before revision

“From the measured modal spectra, we see that, if the beam’s narrower-width section is in the stronger turbulence (smaller r_0), it suffers less turbulence-induced mode coupling than the beam wider section.”

-Sentence after revision

“As shown in the measured modal spectra, the beam suffers less turbulence-induced modal coupling when its narrower-width section is in the stronger turbulence (smaller r_0).”

Furthermore, several examples of truncated figure captions in the main text of the manuscript are presented below (and some of the truncated text in the captions has been moved to the body of the main text):

- The caption of Fig. 1:

-Caption of Fig. 1 before revision

“Fig. 1. (a) A general scheme of using forward-propagating optical beams for probing turbulence along a path. At the Rx, beam-turbulence interactions are measured for retrieving turbulence information. (b) One prior turbulence probing technique by transmitting two probe beams from two separate sources and detecting them at a multi-element Rx aperture array. By changing the angle at which the two beams cross each other, they overlap at different z locations. At these locations, there is a commonality in how the light is affected by the surrounding turbulence. By detecting the difference in the turbulence-induced distortions on the two beams for different angles, the turbulence strength along the z -axis can be measured. (c) Our proposed approach designs and sequentially transmits multiple longitudinally structured beams, each having its narrow beam width at a different z along the propagation path, using a single pair of Tx/Rx. Our longitudinally structured beams are superpositions of multiple $BG_{\ell=0,k_z}$ modes with different longitudinal wavenumbers, k_z , forming a comb in the space-frequency domain. Proper choice of the superposition coefficients dictates the range distribution (along z) of the on-axis intensity pattern of the beam. The three numerically generated transverse (y)-longitudinal (z) intensity distributions exemplify three different superpositions, which result in beams having significant on-axis intensity and narrower beam width only in limited regions of choice. At the Rx, based on measured modal coupling from $\ell = 0$ to $\ell \neq 0$ orders, the distributed turbulence strength along the propagation path is extracted.”

-Caption of Fig. 1 after revision

“Fig. 1. (a) A general scheme for using forward-propagating optical beams to probe turbulence along a path. At the Rx, beam-turbulence interactions are measured to retrieve turbulence information. (b) One prior turbulence probing technique transmits two beams from two separate

sources, crosses them at different distances, and detects them using a multi-element Rx aperture array. (c) Our approach designs and sequentially transmits multiple longitudinally structured beams. The three numerically generated transverse (y)-longitudinal (z) intensity distributions exemplify three different beams with narrower beam widths only in limited regions of choice. At the Rx, based on measured modal coupling from $\ell = 0$ to $\ell \neq 0$ orders, the distributed turbulence strength along the propagation path is extracted.

- The caption of Fig. 3:

-Caption of Fig. 3 before revision

“Fig. 3. Simulation results for probing a “Gaussian-shaped” turbulence distribution, having the peak turbulence strength located at different distances, z , in a 10-km path (e.g., $z=0, 2.5, 5, 7.5, 10$ km). (a) Received beam profiles of Beam 1, 20, and 40 for probing the “Gaussian-shaped” turbulence distribution with the peak located at the middle of the path ($z=5$ km) under one turbulence realization. (b) Average $P(\ell = 0)$ for the 40 beams under 200 turbulence realizations for different “Gaussian-shaped” turbulence distributions. The bars show the standard deviation of the simulation results for each beam. (c) The original turbulence strength distribution and its simulated probed values using the 40 probe beams (in both logarithmic and linear ordinate scale).”

-Caption of Fig. 3 after revision

“Fig. 3. Simulation results for probing “Gaussian-shaped” turbulence distributions, each with the peak turbulence strength located at a different distance, z , in a 10-km path. (a) Received beam profiles of Beam 1, 20, and 40 for probing the turbulence distribution with its peak strength located at $z = 5$ km under one turbulence realization. (b) Average $P(\ell = 0)$ for the probe beams under 200 turbulence realizations. The bars show the standard deviation of the results. (c) The original turbulence distribution and its simulated probed values.”

- The caption of Fig. 5:

-Caption of Fig. 5 before revision

“Fig. 5. (a) Experimental setup for probing two emulated turbulence regions using longitudinally structured beams. Turbulence in each region is emulated by placing a rotatable thin phase plate in the middle of the region. The phase plates are fabricated to have different Fried parameter r_0 to emulate different turbulence strengths (smaller r_0 means stronger turbulence). At Tx, we load different patterns on a spatial light modulator (SLM) to shape the wavefront of a Gaussian beam for the generation of two longitudinally structured beams (i.e., Beam 1 and Beam 2), having their narrow beam widths at $0 < z < 0.3\text{m}$ and $0.3\text{m} < z < 0.6\text{m}$, respectively. In the experiment, we set $Q = 0.999997 \times k$ and $N = 2$. At Rx, we use the off-axis holography approach to measure the spatial amplitude and phase of the beams and calculate their modal power coupling induced by turbulence. (b-c) Simulated and experimentally measured intensity profiles of Beam 1 and 2 at different propagation distances. (d) Simulated and experimentally measured beam widths for Beam 1 and 2. (e) Experimentally measured beam profiles and modal spectrum of Beam 1 and 2 under one turbulence realization for 4 different cases where turbulence regions 1 and 2 have different Fried parameters r_0 .”

-Caption of Fig. 5 after revision

“Fig. 5. (a) Experimental setup for probing two emulated turbulence regions using two longitudinally structured beams (see “Methods” for more details). In the experiment, we set $Q = 0.999997 \times k$ and $N = 2$ to generate the two beams (Beam 1 and 2). They have narrow beam

widths at $0 < z < 0.3\text{m}$ and $0.3\text{m} < z < 0.6\text{m}$, respectively. Simulated and experimentally measured (b-c) intensity profiles and (d) beam widths of Beam 1 and 2 at different propagation distances. (e) Experimentally measured profiles and modal spectra of Beam 1 and 2 under one turbulence realization for four different turbulence cases, where regions 1 and 2 have different r_0 .

Comment 1 (b): Some sentences are incorrect “the atmosphere can seriously affect various applications?” seriously? Maybe turbulence?

Response 1 (b): We thank the reviewer for pointing out some incorrect statements. We have made various corrections (as delineated in other comments of our response). For example, we have modified the sentence in paragraph 1 on page 1 based on the reviewer’s correct comment here:

“Moreover, the atmospheric turbulence can seriously affect various applications.”

Comment 1 (c): The technique is called LIDAR, this is an acronym.

Response 1 (c): We have corrected the word “lidar” in paragraph 3 on page 2:

“One technique is based on LIDAR [5,8–10], which detects a transmitted pulse that is backscattered by air.”

Comment 1 (d): Caption to Fig 1 must be shortened and a brief description must be moved to the text.

Response 1 (d): We have shortened the caption of Fig. 1 and moved some descriptions to the main text:

-Caption of Fig. 1 before revision

“Fig. 1. (a) A general scheme of using forward-propagating optical beams for probing turbulence along a path. At the Rx, beam-turbulence interactions are measured for retrieving turbulence information. (b) One prior turbulence probing technique by transmitting two probe beams from two separate sources and detecting them at a multi-element Rx aperture array. By changing the angle at which the two beams cross each other, they overlap at different z locations. At these locations, there is a commonality in how the light is affected by the surrounding turbulence. By detecting the difference in the turbulence-induced distortions on the two beams for different angles, the turbulence strength along the z -axis can be measured. (c) Our proposed approach designs and sequentially transmits multiple longitudinally structured beams, each having its narrow beam width at a different z along the propagation path, using a single pair of Tx/Rx. Our longitudinally structured beams are superpositions of multiple $BG_{\ell=0, k_z}$ modes with different longitudinal wavenumbers, k_z , forming a comb in the space-frequency domain. Proper choice of the superposition coefficients dictates the range distribution (along z) of the on-axis intensity pattern of the beam. The three numerically generated transverse (y)-longitudinal (z) intensity distributions exemplify three different superpositions, which result in beams having significant on-axis intensity and narrower beam width only in limited regions of choice. At the Rx, based on measured modal coupling from $\ell = 0$ to $\ell \neq 0$ orders, the distributed turbulence strength along the propagation path is extracted.”

-Caption of Fig. 1 after revision:

“Fig. 1. (a) A general scheme for using forward-propagating optical beams to probe turbulence along a path. At the Rx, beam-turbulence interactions are measured to retrieve turbulence information. (b) One prior turbulence probing technique transmits two beams from two separate

sources, crosses them at different distances, and detects them using a multi-element Rx aperture array. (c) Our approach designs and sequentially transmits multiple longitudinally structured beams. The three numerically generated transverse (y)-longitudinal (z) intensity distributions exemplify three different beams with narrower beam widths only in limited regions of choice. At the Rx, based on measured modal coupling from $\ell = 0$ to $\ell \neq 0$ orders, the distributed turbulence strength along the propagation path is extracted.”

-Some descriptions for Fig. 1 have been moved to the main text in paragraph 3 on page 7:

“Figure 1(b) shows a prior turbulence probing approach using two crossing beams. By changing the angle of the two beams, they overlap at different z locations. At these locations, there is a commonality in how the light is affected by the turbulence. By detecting the difference in the turbulence-induced distortions for different angles, the turbulence strength along z can be measured. Figure 1(c) shows the concept of our approach using a single Tx/Rx pair and multiple longitudinally structured beams, each with its narrow width at a different position along the path. Our longitudinally structured beams are superpositions of multiple $BG_{\ell=0,k_z}$ modes with different k_z . Proper choice of the superposition coefficients dictates the distribution (along z) of the on-axis intensity pattern of the beam, which also corresponds to different beam widths along z.”

Comment 1 (e): Lines 96-102 too long sentence!

Response 1 (e): We thank the reviewer for pointing it out. We have shortened the sentence:

“Each beam is a superposition of multiple $\ell = 0$ order $BG_{\ell=0,k_z}$ modes with different longitudinal wavenumbers, k_z [26]. BG_{ℓ,k_z} modes have two spatial indices: (i) ℓ is the number of 2π phase shifts in the azimuthal direction (i.e., the beam’s OAM value), and (ii) k_z is related to the radial wavenumber k_r , which determines the radial ring spacings in the intensity profile [27, 28]. k_z and k_r satisfy $k_z^2 + k_r^2 = \left(\frac{2\pi}{\lambda}\right)^2$, where λ is the wavelength [27, 28].”

Comment 1 (f): Formula in 101 is incorrect, check brackets.

Response 1 (e) & (f): The reviewer is correct. We have corrected the formula:

“Each beam is a superposition of multiple $\ell = 0$ order $BG_{\ell=0,k_z}$ modes with different longitudinal wavenumbers, k_z [26]. BG_{ℓ,k_z} modes have two spatial indices: (i) ℓ is the number of 2π phase shifts in the azimuthal direction (i.e., the beam’s OAM value), and (ii) k_z is related to the radial wavenumber k_r , which determines the radial ring spacings in the intensity profile [27, 28]. k_z and k_r satisfy $k_z^2 + k_r^2 = \left(\frac{2\pi}{\lambda}\right)^2$, where λ is the wavelength [27, 28].”

Comment 2: My main concern is related to the comparison of theoretical results with the experimental data. I do not believe that at the distance of 0.6 m we have a fully developed turbulence, otherwise that we have Kolmogorov's statistics. But the main results of the theory are based on the Kolmogorov's similarity theory. is it possible to provide a proof, that turbulence in the lab experiment was fully developed?

Response 2: We feel that the reviewer has raised an important issue. To address this valuable point, we have: (a) experimentally examined our emulated turbulence by measuring the Strehl ratio and the power fluctuation for a Gaussian beam, (b) added a new figure and corresponding descriptions for our new

measurements in the *Supplementary Information*, (c) added new sentences to explain and discuss our new measurements in the main text of the manuscript, and (d) added several relevant new references. Through these additions and modifications, we hope that the reader will have a better understanding of the extent to which our experimental emulation of turbulence after propagation through a phase plate might be a reasonable representation of Kolmogorov statistics.

Specifically, we have added sentences in the *Methods* section of the main text on page 20 of the revised manuscript:

“We divide the 0.6-m path into two equal-length regions and emulate turbulence effects by placing a rotatable thin phase plate in the middle of each turbulence region (i.e., 0.3-m regions with at least 0.15-m free-space propagation after the plate). The phase plates are fabricated with a pseudo-random phase distribution obeying Kolmogorov spectrum statistics [1, 24, 57, 58]. They are characterized by different Fried parameters r_0 (e.g., 0.4, 1, and 3 mm) to emulate different turbulence strengths; smaller r_0 corresponds to stronger turbulence [1, 57]. In order to emulate different turbulence realizations, we rotate the phase plates so that the beam passes through different representations of turbulence [57].

We examine if our emulation of turbulence after ≥ 0.15 -m beam propagation after the phase plate can be considered close to Kolmogorov statistics by measuring the Strehl ratio (SR) for a Gaussian beam [1] (see “Supplementary information” Fig. S7). We measure SR values for different phase plates under various link lengths (from 0.3 to 0.6 m). Our results show that the measured values are close to the theoretical values with $< 8\%$ relative errors. These results indicate that the emulated turbulence exhibits a reasonable representation of a Kolmogorov power spectrum [57, 59, 60]. In addition, we measure the statistics of the power fluctuation of the received beam passing through a phase plate with $r_0 = 1$ mm. The probability density function of measured fluctuations follows a lognormal model [57] with a correlation coefficient $R > 0.96$ to Kolmogorov statistics for various link lengths and propagation after phase plates. Moreover, the scintillation index is found to be larger for a longer link, which indicates larger intensity fluctuations caused by the emulated turbulence [61]. If the receiver is placed right after the phase plate, we note that the received beam has negligible intensity fluctuations and the effective turbulence strength approaches zero [61]; intensity fluctuations would thus arise only after the beam has propagated some distance after the plate [61].

Previously, phase plates have been utilized in various laboratory experiments for emulating turbulence with Kolmogorov statistics in a relatively short path [24, 57, 59, 60]. Importantly, the turbulent path emulated by phase plates in the laboratory can correspond to a much longer path [1, 62]. Turbulence parameters of a longer path can be scaled from a shorter lab path given that the two systems have a similar Fresnel number [62]

$$F = a^2 / (\lambda L) \tag{16}$$

where a denotes the radius of the source aperture and L is the link length. Based on our calculation, each turbulence region emulated in our experiment corresponds to a 5-km path segment with $C_n^2 = 1.3 \times 10^{-14} \text{ m}^{-2/3}$, $3.2 \times 10^{-15} \text{ m}^{-2/3}$, and $4.3 \times 10^{-16} \text{ m}^{-2/3}$ for phase plates with $r_0 = 0.4$ mm, 1.0 mm, and 3.0 mm, respectively.”

We have also added the corresponding new references in the main text:

[57] Y. Ren, H. Huang, G. Xie, N. Ahmed, Y. Yan, B. I. Erkmen, N. Chandrasekaran, M. P. J. Lavery, N. K. Steinhoff, M. Tur, S. Dolinar, M. Neifeld, M. J. Padgett, R. W. Boyd, J. H. Shapiro,

and A. E. Willner, "Atmospheric turbulence effects on the performance of a free space optical link employing orbital angular momentum multiplexing," *Opt. Lett.* 38, 4062-4065 (2013).

[58] R. Rampy, D. Gavel, D. Dillon, and S. Thomas, "Production of phase screens for simulation of atmospheric turbulence," *Applied Optics* 51, 8769-8778 (2012).

[59] B. Rodenburg, M. Mirhosseini, M. Malik, O. S. Magaña-Loaiza, M. Yanakas, L. Maher, N. K. Steinhoff, G. A. Tyler, and R. W. Boyd, "Simulating thick atmospheric turbulence in the lab with application to orbital angular momentum communication," *New Journal of Physics* 16, 033020 (2014).

[60] A. Klug, C. Peters, and A. Forbes, "Robust structured light in atmospheric turbulence," *Advanced Photonics* 5, 016006 (2023).

[61] L. C. Andrews, R. L. Phillips, and A. R. Weeks, "Propagation of a Gaussian-beam wave through a random phase screen," *Waves in Random Media* 7, 229 (1997).

[62] J. D. Phillips, M. E. Goda, and J. Schmidt, "Atmospheric turbulence simulation using liquid crystal spatial light modulators," In *Advanced Wavefront Control: Methods, Devices, and Applications III*, vol. 5894, pp. 57-67. SPIE, 2005.

Furthermore, we have added in the *Supplementary Information* the following new experimental results, a new figure (Fig. S7), and corresponding explanations:

"As shown in Fig. S7 (a), we measure the Strehl ratio (SR) [1] for different phase plates (different r_0) with various path lengths (i.e., $L=0.3$ and 0.6 m). To measure each data point, we place the corresponding phase plate in the middle of the path (i.e., $z = L/2$) and propagate a Gaussian beam through it. The width of the Gaussian beam is $D=3.5$ mm. At the receiver, we measure SR values and compare them to the theoretical ones, which can be expressed as $[1 + (D/r_0)^{5/3}]^{-6/5}$ [1, 24]. Our results show that the measured values are close to the theoretical ones with $<8\%$ relative errors. In Fig. S7 (b), we measure the received power fluctuation of the Gaussian beam for the phase plate with $r_0 = 1$ mm. For these measurements, the receiver aperture diameter is ~ 1 mm. Our results show that the probability density function of measured power fluctuations follows a lognormal model for each path length. The correlation coefficient R between the distribution and its lognormal fitting curve is >0.96 . We also calculate the scintillation index σ_I^2 [1] and find that it increases from 0.093 to 0.161 when L changes from 0.3 to 0.6 m."

Fig. S7. (a) Measured SR for different phase plates (different r_0) with different path lengths (different L). The beam width of the Gaussian beam is $D=3.5$ mm. (b) Measured probability density function of power fluctuations of the received beam for the phase plate with $r_0 = 1$ mm.

Comment 3: Clear conclusions (and conclusions in general) are absent. Please, add.

Response 3: The reviewer makes a good point for the reader. We have added a *Summary* section on page 20 of the revised main text of the manuscript:

“Summary:

In this paper, we propose and demonstrate an approach using sequentially transmitted longitudinally structured beams to probe turbulence strength along a propagation path. Each beam is a superposition of multiple BG modes and is designed to have a smaller beam width at a different longitudinal region. Since turbulence can affect a wider beam more than a narrower beam, we extract the turbulence-strength distribution after measuring the turbulence-induced modal coupling for different beams at the receiver. Our simulation shows (i) relatively uniform probing errors (~ 0.1 to ~ 0.3 dB) of our approach for a 10-km turbulence path with up to a 30-dB difference in C_n^2 , and (ii) a trade-off between probing resolution and transmitter aperture size. We experimentally demonstrate our approach for probing two emulated turbulence regions with up to a ~ 15 -dB turbulence strength variation. The experimental results show < 0.8 -dB probing errors. Compared to previous methods, our approach has the potential to (i) support longer distances or (ii) achieve fairly uniform performance along the path using smaller transceivers.”

Comment 4: In the abstract authors claimed "Here, we explore turbulence probing utilizing multiple sequentially transmitted longitudinally structured beams between a single transmitter/receiver aperture pair." The refractive index structure parameter measurements are available in the literature data. is it possible to provide a comparison with the available experimental data in atmosphere?

Response 4: The reviewer makes an excellent and valuable comment. We have added the following sentences in paragraph 1 on page 15 of the revised manuscript:

“We also simulate a C_n^2 profile based on the Hufnagel-Valley (H-V) model, which has been widely used to describe turbulence strength at different altitudes [50]. In addition, we simulate our approach and compare the results to previously published experimental data of C_n^2 at different altitudes measured by radiometers [51] (see “Supplementary Information” Fig. S4). The simulated probing error compared to the H-V model and the experimental data published in the literature [51] is ~8% and ~16%, respectively.”

We have added the corresponding new reference in the main text:

[51] Brian E. Vyhnaek, "Path profiles of C_n^2 derived from radiometer temperature measurements and geometrical ray tracing," In Free-Space Laser Communication and Atmospheric Propagation XXIX, vol. 10096, pp. 386-396. SPIE, 2017.

Furthermore, we have added in the *Supplementary Information* the following new simulation results, two new subfigures (Fig. S4 (g) and (h)), corresponding explanations, and a new reference:

“Figure. S4 (g) shows the simulation results for an atmospheric turbulence profile based on the Hufnagel-Valley (H-V) model. We also simulate our approach and compare it to experimental measurements of C_n^2 by radiometers at different altitudes in the literature [8], as shown in Fig. S5 (h). Our results show that (i) the probed turbulence distribution over altitudes has a similar trend as the original turbulence and (ii) the simulated probing error compared to the H-V model and the experimental data in the literature is ~8% and ~16%, respectively.”

We have also added the corresponding new reference in the *Supplementary Information*:

[8] Brian E. Vyhnaek, "Path profiles of Cn^2 derived from radiometer temperature measurements and geometrical ray tracing," In *Free-Space Laser Communication and Atmospheric Propagation XXIX*, vol. 10096, pp. 386-396. SPIE, 2017.

Comment 5: Furthermore, it is not clear what is the role of the orbital angular momentum in calculations, because the formula 10 is the approximate formula. Is it possible to add the discussion about OAM and how it will affect the results and accuracy?

Response 5: We thank the reviewer for this insightful comment. We have added the following sentences in paragraph 2 on page 8 of the main text:

"Based on Eqs. (1-2), the normalized average received power remaining on the $\ell = 0$ order for beam i is approximated by [24]:

$$P_i(\ell = 0) \approx (I_0(\beta_i) + I_1(\beta_i)) \exp(-\beta_i) \quad (10)$$

Therefore, after measuring $P_i(\ell = 0)$ for each transmitted beam at the Rx and extracting the corresponding β_i based on Eq. (10), we can solve the M equations of Eq. (9) for the $\{C_{n,j}^2, j = 1, \dots, M\}$ of the different regions.

We note that previous studies have reported a more general analytic expression to calculate normalized average received power on different OAM orders ($P(\ell)$) [24]:

$$P(\ell) \approx \frac{\beta^\ell \cdot {}_2F_2\left(\frac{1}{2} + \ell, 1 + \ell; 2 + \ell, 1 + 2\ell, -2\beta\right)}{2^\ell \Gamma(2 + \ell)} \quad (11)$$

where $\Gamma(\cdot)$ is the gamma function and ${}_2F_2$ is the generalized hypergeometric function. Eq. (10) is reduced from Eq. (11) for the case of $\ell = 0$ [24]. Both Eq. (10) and Eq. (11) are approximated formulas that only consider the low-order turbulence aberrations (e.g., tip and tilt) [24]. This approximation is based on that the contribution to OAM modal coupling of higher-order aberrations diminishes rapidly and can be orders of magnitude smaller [24, 39]. It has been shown that Eq. (11) generally underestimates $P(\ell)$, but it is in relatively close agreement with the exact values for $\ell = 0, 1$ [24].

In order to measure the normalized $P(\ell = 0)$, we perform spatial modal decomposition and normalize the power among OAM orders ranging from $\ell = -10$ to $\ell = +10$ [24, 27]. If a larger number of OAM orders are taken into account during the measurements, a more accurate normalized value of $P(\ell = 0)$ might be obtained. Moreover, it is also possible to utilize the power on many other $\ell \neq 0$ orders as signatures for probing based on Eq. (11). Therefore, more information on the modal coupling can be utilized for turbulence probing with potentially higher accuracy."

We have also added the corresponding new reference in the main text:

[39] Robert J. Noll, "Zernike polynomials and atmospheric turbulence," *J. Opt. Soc. Am.* 66, 207-211 (1976).

Reviewer 2

The authors treated the problem as to probing the distribution of turbulence strength along a path. The basic idea of this manuscript is interesting and the work is important. The approach suggested by the authors may be useful for some application scenarios. However, there are some drawbacks in the current version of the manuscript that deserve careful consideration.

Comment 1: As for Eq. (2), is the parameter D the beam width at the transmitting plane or the receiving plane? If D is the beam width at the receiving plane, it will depend on the turbulence strength distribution due to the turbulence-induced beam spreading. In this sense, D should not be independent of r_0 . If D is the beam width at the transmitting plane, the authors should note that the beam wave filed at the transmitting plane should be coherent and hence D is actually the beam width of a coherent beam. For Eq. (4), D_j with $j = 2, 3, \dots, M$ therein should be considered as the beam width of a partially coherent beam at the input plane of a segment because the beam becomes partially coherent during the propagation in turbulence. Eq. (4) is based on Eq. (2). For a partially coherent beam, whether Eqs. (1) and (2) are valid or not? If not, the theoretical foundation of the manuscript is problematic. It seems that Ref. [24] did not consider the partial coherence issue.

Response 1: We thank the reviewer for raising a valuable issue. To address this important point, we have (a) performed new simulations and calculations to examine the use of Eqs. (1-2) for our inhomogeneous case, (b) added a new figure and corresponding descriptions for our new simulation and calculation results in the *Supplementary Information*, (c) added new sentences to explain our new results in the main text, and (d) added relevant new references. Through these additions and modifications, we hope that the reader will have a better understanding of the extent to which Eqs. (1-2) may provide an approximation in inhomogeneous turbulence cases where the input beam is partially coherent for turbulence regions j ($j \geq 2$).

Specifically, we have added the following sentences in paragraph 1 on page 8 of the main text:

“At the Tx, M longitudinally structured beams are sequentially transmitted, in which beam i has its narrow width in region i . Following Eq. (4), the β for beam i can be approximated by:

$$\beta_i \approx 1.8025 \sum_{j=1}^M \left\{ (D_{i,j})^{\frac{5}{3}} [0.423k^2 C_{n,j}^2 \Delta z] \right\} \quad (8)$$

*where $D_{i,j}$ is the width of beam i when it is in turbulence region j , and $\Delta z = L/M$ is the length of each region. **We note that the beam width is not a constant value and can change along z during beam propagation within each region [27]. In Eq. (8), we use a constant beam width value $D_{i,j}$ as an approximation to calculate the turbulence effect for each region. For beam i , $D_{i,j}$ is defined as the average value of the beam width along z within turbulence region j .**”*

In addition, we have added the following paragraphs on page 9 of the main text:

“We note that our approach applies Eqs. (1-2) (i.e., equations for a homogeneous medium in Ref. [24]) to an inhomogeneous turbulence scenario comprising multiple longitudinal segments. In Ref. [24], homogeneous turbulence (only a single segment) is considered and the input beam is a single-mode coherent beam. However, we consider inhomogeneous turbulence cases (multiple segments) in our paper. As a result, the input beam for turbulence regions j ($j \geq 2$) can be only partially spatially coherent [40]. This is because the beam can contain many spatial modes after propagating through turbulence, which can generally decrease the spatial coherence of the beam [41, 42]. Thus, it is valuable to study whether Eqs. (1-2) are applicable to each beam i propagating through turbulence regions j ($j \geq 2$).”

In “Supplementary Information” Section 1, we use simulations to help determine whether Eqs. (1-2) can provide reasonable estimations in several cases of inhomogeneous turbulence distributions each comprising three regions. For each case, we simulate $P_{ij}(\ell = 0)$ for beam i

at the end of region j and compare it to the theoretically calculated value based on Eqs. (1-2). Our results show that $P_{i,j}(\ell = 0)$ become smaller after more turbulence regions (e.g., j becomes larger), which might be due to the more significant accumulation of turbulence-induced modal coupling [42]. The calculated results are in relatively close agreement with the simulated results with <5% errors. Our simulation results seem to indicate that Eqs. (1-2) might provide an approximation of turbulence-induced modal coupling results for turbulence regions j ($j \geq 2$) in inhomogeneous turbulence cases. However, a more comprehensive and rigorous theoretical analysis may be beneficial in the future in order to further investigate modal coupling effects on partially coherent beams propagating in various inhomogeneous turbulence cases [40-43].”

We have also added the corresponding new references in the main text:

[40] G. Gbur, "Partially coherent beam propagation in atmospheric turbulence," *JOSA A* 31, 2038-2045 (2014).

[41] T. Shirai, A. Dogariu, and E. Wolf, "Mode analysis of spreading of partially coherent beams propagating through atmospheric turbulence," *JOSA A* 20, 1094-1102 (2003).

[42] C. Schwartz and A. Dogariu, "Mode coupling approach to beam propagation in atmospheric turbulence," *JOSA A* 23, 329-338 (2006).

[43] J. Zhou, J. Zong, and D. Liu. "Coupled mode theory for orbital angular momentum modes transmission in the presence of atmospheric turbulence," *Optics Express* 23, 31964-31976 (2015).

Furthermore, we have added in the *Supplementary Information* the following new simulation and calculation results, as well as a new figure (Fig. S1) and corresponding explanations:

“To help examine the use of Eqs. (1-2) for inhomogeneous turbulence scenarios, we simulate beam propagation through different turbulence distributions each comprising three turbulence regions. As shown in Fig. S1, we simulate $P_{i,j}(\ell = 0)$ for beam i at the output plane of region j using spatial modal decomposition (see “Methods” in the main text). We also calculate $P_{i,j}(\ell = 0)$ based on Eqs. (1-2) and compare the results to the simulated ones. The calculation methods are as follows:

- Calculation based on Eqs. (1-2):

Eqs. (1-2) describe how much power will be coupled from the $\ell = 0$ order to other $\ell \neq 0$ orders due to turbulence-induced modal coupling [24]. Based on Eqs. (1-2), we can calculate the normalized average power remaining on the $\ell = 0$ order ($P(\ell = 0)$) after the beam propagates through a turbulence region. In inhomogeneous turbulence cases with multiple uniform turbulence regions, we define $P_{i,j}(\ell = 0)$ as the power remaining on the $\ell = 0$ order for beam i at the end of region j . Due to the accumulated effects of multiple turbulence regions, there can be increased modal power coupling during beam propagation and the resulting $P_{i,j}(\ell = 0)$ will decrease with a larger j .

For example, after propagation through the first turbulence region, $P_{i,1}(\ell = 0)$ can be calculated based on Eqs. (1-2) [24], as follows:

$$P_{i,1}(\ell = 0) = \left(I_0(\beta_{i,1}) + I_1(\beta_{i,1}) \right) \exp(-\beta_{i,1}) \quad (S1)$$

and

$$\beta_{i,1} \approx 1.8025(D_{i,1})^{\frac{5}{3}}[0.423k^2C_{n,1}^2\Delta z] \quad (S2)$$

where $D_{i,1}$ is the beam width of beam i in the first region, and $C_{n,1}^2$ is the turbulence strength of the first region.

After propagating through the next turbulence regions, the beam experiences stronger modal coupling, and more power on the $\ell = 0$ order will be coupled to $\ell \neq 0$ orders. As a result, the relative power remaining on the $\ell = 0$ will decrease and the $P_{i,j}(\ell = 0)$ is calculated as follows [24]:

$$P_{i,j}(\ell = 0) \approx P_{i,0}(\ell = 0) \prod_{m=1}^j (I_0(\beta_{i,m}) + I_1(\beta_{i,m})) \exp(-\beta_{i,m}) \quad (S3)$$

and

$$\beta_{i,m} \approx 1.8025(D_{i,m})^{\frac{5}{3}}[0.423k^2C_{n,m}^2\Delta z] \quad (S4)$$

where the value of $P_{i,0}(\ell = 0)$ is 1 at the transmitter, $D_{i,m}$ is the beam width in region m , and $C_{n,m}^2$ is the turbulence strength of region m .

- Simulation and calculation results:

As shown in Fig. S1, our simulation and calculation results show that $P(\ell = 0)$ becomes smaller after the beam propagates through more turbulence regions. This might be because larger accumulated turbulence effects cause more power to be coupled from the $\ell = 0$ order. Moreover, a stronger turbulence region causes a greater decrease in the $P(\ell = 0)$ for each beam due to the stronger modal coupling effect in this region. The calculated results are in relative agreement with the simulated results and show <5% average relative errors."

Fig. S1. Simulated and calculated $P(\ell = 0)$ values for each probe beam under three different turbulence strength distribution cases each containing three turbulence regions. For each case, $P(\ell = 0)$ values are simulated and calculated at the end of each region."

Comment 2 (a): As an optical beam propagates through atmospheric turbulence, the beam width may not be identical to that of the vacuum propagation case. Indeed, atmospheric turbulence may change the beam width at a given propagation distance. However, in “Beam width calculation” of the “Method” section, the authors did not mention the turbulence-induced beam width variation. In my opinion, the authors should analyze how the turbulence-induced beam width variation affects the accuracy of their approach.

Comment 2 (b): Because of the existence of the turbulence-induced beam width variation, the design of the location of intensity-higher region should consider the effects of atmospheric turbulence. I suggest that the authors address this issue rigorously.

Response 2 (a) and (b): The reviewer makes two excellent comments. We feel that both comments are related to the effects of turbulence-induced beam width variations, and therefore we treat them together below. Specifically, the turbulence-induced beam width variations may: (a) change the beam width and make it non-identical to our designed one in the vacuum, and (b) affect the designed location of the intensity-higher region (smaller-beam-width region) of our probe beams. To address these issues, we have: (i) performed new simulations to investigate the effects of turbulence-induced beam width variations on our approach, (ii) added two new figures and corresponding descriptions for our new simulation results in the *Supplementary Information*, (iii) added new explanations and discussions in the main text, and (iv) added several relevant new references. Through these additions and modifications, we hope that the reader will have a better understanding of the effects of turbulence-induced beam width variations on our approach.

Specifically, we have added the following paragraphs on page 10 of the main text, including

- a paragraph about the issues that might be caused by turbulence-induced beam width variations:

“In our approach, we utilize the designed beam width in the vacuum (without turbulence) to form equations for retrieving longitudinal turbulence strengths. However, turbulence can cause beam width variations at a given propagation distance [1, 44, 45]. Therefore, the actual beam width in turbulence may not be identical to that of our designed beam. Moreover, beam width variations may also affect the location of the intensity-higher region (the smaller-beam-width region) [46, 47]. In “Supplementary Information” Section 2, we simulate turbulence-induced beam width variations under several turbulence distributions and investigate how such variations would affect our probing approach.”

- a paragraph about new simulation results for turbulence-induced beam width variations:

“Our results show two effects caused by turbulence-induced beam width variations: (i) the beam width can be affected by turbulence and become larger than the designed beam width in the vacuum (i.e., beam spreading) [44, 45], and (ii) the location of the smaller-beam-width region can be shifted closer to the transmitter under turbulence (i.e., location shift) [46, 47] (see “Supplementary Information” Fig. S2). Moreover, these two effects are related to both the turbulence distribution and the design of the probe beam. Specifically, our results show that (i) the beam spreading tends to be more significant if stronger turbulence is closer to the transmitter [48], and (ii) the location shift is larger for a probe beam with its smaller-beam-width region located further from the transmitter. (see “Supplementary Information” Section 2 for more explanations).”

- a paragraph about new simulation results for the effects of beam width variations on our approach:

“Furthermore, we compare the turbulence probing performance when using the designed beam width in the vacuum or the average beam width in turbulence (see “Supplementary Information” Fig. S3). The results show that the probing error is ~2% larger if the beam width changes in turbulence are not considered. We also simulate other turbulence distributions with different numbers of regions and find similar effects of beam width variations on our approach.”

In these specific cases, our simulation results seem to indicate that our approach may suffer a relatively small decrease in the probing accuracy if we do not consider the beam width changes in turbulence. In general, the beam width variation can become more significant for a beam propagating through a longer and stronger turbulent path [1, 44]. Therefore, it may have a greater effect on our approach. We note that a more rigorous theoretical study may be beneficial in the future in order to: (a) examine the extent that beam-width variations affect the accuracy of our approach, and (b) help optimize the design of the probe beams for better performance in various turbulence cases [46, 47].

We have also added the corresponding new references in the main text:

[44] H. T. Yura, "Short-term average optical-beam spread in a turbulent medium," J. Opt. Soc. Am. 63, 567-572 (1973).

[45] C. Y. Young, Y. V. Gilchrest, and B. R. Macon. "Turbulence induced beam spreading of higher order mode optical waves," Optical Engineering 41, 1097-1103 (2002).

[46] J. C. Ricklin, W. B. Miller, and L. C. Andrews. "Optical turbulence effects on focused laser beams: new results," In Optics in Atmospheric Propagation and Random Phenomena, vol. 2312, pp. 145-154. SPIE, 1994.

[47] J. C. Ricklin, W. B. Miller, and L. C. Andrews. "Effective beam parameters and the turbulent beam waist for convergent Gaussian beams," Applied Optics 34, 7059-7065 (1995).

[48] Y. K. Chahine, S. A. Tedder, B. E. Vyhnalek, and A. C. Wroblewski. "Beam propagation through atmospheric turbulence using an altitude-dependent structure profile with non-uniformly distributed phase screens," In Free-Space Laser Communications XXXII, vol. 11272, pp. 263-277. SPIE, 2020.

Furthermore, we have added in the *Supplementary Information* the following new simulation results, two new figures (Figs. S2-S3), new corresponding explanations, and new references:

"We simulate the turbulence-induced beam width variation and its effects on our probing approach. As shown in Fig. S2, we simulate different turbulence distribution cases each containing three turbulence regions. We design three sequentially transmitted probe beams to probe these distributions along a 10-km path. For each probe beam, we first simulate its propagation in the vacuum (without turbulence) and calculate its beam width at various distances [31] (see "Methods" in the main text for beam width calculation). We can find that the beam width is smaller in one specific region (i.e., smaller-beam-width region) as we designed.

Next, we simulate beam propagation through different turbulence distributions and calculate the beam width at different distances under 200 turbulence realizations. The orange shades in Fig. S2 show the range of beam width variations induced by turbulence, which is larger for a longer propagation distance due to stronger accumulated turbulence effects [1, 2]. When the stronger turbulence region is closer to the transmitter, the beam width has larger variations (e.g., comparing Case 1 to Case 3). This might be due to that severe turbulence distortion near the transmitter causes stronger beam variations after a longer-distance propagation [3]. We also simulate the average beam width in turbulence and compare it to the designed beam width in the vacuum. The results show that the average beam width in turbulence is larger than that in the vacuum due to turbulence-induced beam spreading [4].

Besides the beam spreading, turbulence-induced beam width variations can also change the longitudinal location of the smaller-beam-width region [5,6]. An example is indicated in Fig. S2 (a3), where the location of the smaller-beam-width region will be shifted closer to the transmitter under turbulence. Previous studies have also shown a similar effect for focused Gaussian beams, in which the beam waist location shifts closer to the transmitter under turbulence [5,6].

We subsequently simulate how the turbulence-induced beam width variations affect our probing approach. Figure. S3 (a) shows an original turbulence distribution. Figure. S3 (b) shows simulated and theoretically calculated $P(\ell = 0)$ values for each probe beam at the receiver. We calculate $P(\ell = 0)$ using the designed beam width in the vacuum and the average beam width in turbulence. Compared to the designed beam width in the vacuum, the calculation results using the average beam width in turbulence are slightly closer ($\sim 3\%$) to the simulation results on average. This might be because (i) the average beam width in turbulence is larger, corresponding to more turbulence-induced modal coupling, and (ii) the designed beam width in the vacuum might underestimate the modal coupling. Subsequently, we use the beam width in the vacuum or the average beam width in turbulence to form equations for retrieving turbulence. As shown in Fig. S3 (c), the probing error is $\sim 2\%$ smaller when using the average beam width in turbulence.”

We have also added the corresponding new references in the *Supplementary Information*:

[2] H. T. Yura, "Short-term average optical-beam spread in a turbulent medium," *J. Opt. Soc. Am.* 63, 567-572 (1973).

[3] Y. K. Chahine, S. A. Tedder, B. E. Vyhnalek, and A. C. Wroblewski. "Beam propagation through atmospheric turbulence using an altitude-dependent structure profile with non-uniformly distributed phase screens," *In Free-Space Laser Communications XXXII*, vol. 11272, pp. 263-277. SPIE, 2020.

[4] C. Y. Young, Y. V. Gilchrest, and B. R. Macon. "Turbulence induced beam spreading of higher order mode optical waves," *Optical Engineering* 41, 1097-1103 (2002).

[5] J. C. Ricklin, W. B. Miller, and L. C. Andrews. "Optical turbulence effects on focused laser beams: new results," *In Optics in Atmospheric Propagation and Random Phenomena*, vol. 2312, pp. 145-154. SPIE, 1994.

[6] J. C. Ricklin, W. B. Miller, and L. C. Andrews. "Effective beam parameters and the turbulent beam waist for convergent Gaussian beams," Applied Optics 34, 7059-7065 (1995).

Comment 3: What does the parameter $k_{r,n}$ in Eq. (5) represent? The authors did not describe the meaning of $k_{r,n}$. However, $k_{\rho,n}$ was explained in the text that follows Eq. (5).

Response 3: We thank the reviewer for catching our typo mistake. We have changed $k_{r,n}$ to $k_{\rho,n}$ in Eq. (5) as pointed out by the red arrow:

“Equation (5) shows the waveform of a longitudinal structured beam consisting of $(2N + 1)$ BG modes all at the same optical frequency ω_0 [26,38]:

$$\Psi(\rho, z, t) = e^{-i\omega_0 t} G(\rho) \sum_{n=-N}^N A_n J_0 \left(k_{\rho,n} \rho \right) e^{ik_{z,n} z} \quad (5)$$

where ρ is the radius in the cylindrical coordinate; $k_{\rho,n}$ and $k_{z,n}$ are transverse and longitudinal wavenumbers, respectively, satisfying $k_{\rho,n}^2 + k_{z,n}^2 = k^2$ ”

REVIEWERS' COMMENTS

Reviewer #1 (Remarks to the Author):

This is a very interesting work and I think it could be potentially significant in the field of aero-optics.

Reviewer #2 (Remarks to the Author):

Thank the authors for the revision. I think the current manuscript can be accepted for publication.

The authors sincerely thank the reviewers for all their significant efforts and insightful comments. We believe firmly that the reviewers have made the paper dramatically better through the peer-review process.

Response to reviewers' comments:

REVIEWER #1

This is a very interesting work and I think it could be potentially significant in the field of aero-optics.

Response: We thank the reviewer for their meaningful encouragement.

REVIEWER #2

Thank the authors for the revision. I think the current manuscript can be accepted for publication.

Response: Again, we thank the reviewer for their meaningful encouragement.